



**Meteorological formation mechanism of regional transport in winter**
**heavy air pollution events in the middle Yangtze River area, China**
Yongqing Bai[1]   Tianliang Zhao[2*]   Yue Zhou[1*]   Jie Xiong[1]   Weiyang Hu[2]   Yao Gu[2]   Lin Liu[1]
Shaofei Kong[3]   Huang Zheng[3]
1. Hubei Key Laboratory for Heavy Rain Monitoring and Warning Research, Institute of Heavy Rain,China
Meteorological Administration, Wuhan 430205, China
2. Key Laboratory for Aerosol-Cloud-Precipitation of China Meteorological Administration, Nanjing University of
Information Science and Technology, Nanjing 210044, China
3. Department of Atmospheric Science, School of Environemtal Studies, China University of Geosciences
(Wuhan), 430074
Correspondence: Tianliang Zhao (josef_zhao@126.com) and Zhou Yue
(zhouyue8510@163.com)
**Abstract:** Anthropogenic emission, meteorological conditions, and regional transport
are the three major factors influencing heavy air pollution in China. The Hunan and
Hubei provinces in the middle Yangtze River region border China's main air pollution
areas, serving as the hub of regional transport of air pollutants. The meteorological
formation mechanism of regional transport of air pollutants on heavy air pollution in
the Hunan and Hubei provinces still remain urgent to be addressed in depth. In this
study, multivariate empirical orthogonal function (MV-EOF) analysis was performed
to objectively select eight typical heavy pollution events in the two provinces that
occured in January 2015–2019. Based on the regional surface environment,
meteorological network data, atmospheric sounding data, ERA-interim reanalysis data,
and a numerical simulation experiment, this study investigated the pattern of regional
transport of air pollutants in the two provinces and its mechanism of regional
meteorological conditions. The results showed that transporting air pollutants mainly
passed through two transport pathways, namely the Nanxiang Basin-Yunmeng Plain
pathway and the Dabie Mountain's Hilly Area-Yunmeng Plain pathway, existing
anomalous near-surface northerly winds in the two provinces and their upstream area
accompanied by southward penetration of a shallow cold layer, all of which jointly
provide a dynamic condition for regional air pollutant transport. The weak cold-air
mass degenerated as it passed through the Hunan–Hubei Plain, causing warm air to

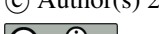



accumulate in the near-surface layer of the downstream area, where winds slowed
down and converged, buffering the air pollutant transport and resulting in pollutants
accumulation; the near-surface atmosphere of the Hunan and Hubei provinces was a
non-stagnant condition (dry intrusion of cold air, anomalous northerly winds, and
positive anomalies of boundary-layer height), which is conducive to the horizontal
transport of air pollutants. However, the mid-high layers, characterized by
temperature inversion and the presence of a "warm lid", had a stable stratification,
inhibiting the diffusion of air pollutants to the upper layers; there is an obvious
longitudinal vertical circulation above the Hunan–Hubei Plain, which results in the
sinking and accumulation of air pollutants, thereby promoting rapid accumulation of
air pollutants in the Hunan and Hubei provinces. In addition, extended empirical
orthogonal function (EEOF) analysis was performed, revealing a quasi-4-d periodic
oscillation pattern of air pollutants transport in the Hunan and Hubei provinces, which
provides a reference for early prediction of its regional transport. The findings are of
practical value in broadening the scientific understanding of the differences in the
formation mechanism of heavy atmospheric pollution between the various regions of
China and promoting environmental and ecological protection of the middle Yangtze
Basin.

## 1 Introduction

At present, China is facing a severe air pollution situation, and in particular,
regional atmospheric environmental problems characterized by the pollution of
inhalable particulate matter ($PM_{10}$) and fine particulate matter ($PM_{2.5}$) are increasingly
prominent, which is the main challenge in China's atmospheric environmental
management (Kan et al., 2012; Zhao et al., 2013; Zhang et al., 2015). Under certain
geographical and meteorological conditions, atmospheric pollutants diffuse and
accumulate on a certain spatial scale. Air pollutants flow freely between cities and
urban agglomerations and undergo trans-boundary transport, characterized by a
combination of regional pollution and compound pollution, with regional joint
prevention and control of pollutant transport becoming the focus of atmospheric
environmental issues in China (Wu et al., 2013b; Miao et al., 2017; Lu et al., 2019a).
Exploring meteorological mechanism of regional air pollutant transport, scientifically
assessing the atmospheric environmental changes, and effectively regulating pollutant
emissions constitute an important topic of atmospheric environmental research



(Cheng et al., 2019; Zhang et al., 2019; Huang et al., 2020).
China's heavy atmospheric pollution often occurs in winter, and the
pollution-promoting meteorological conditions in winter are usually about 40-100%
worse than those in in other seasons (Zhang et al., 2018). The effects of
meteorological conditions on the formation, distribution, maintenance, and change of
aerosol pollution are significant (Tai et al., 2012; Ding et al., 2016; Guo et al., 2019).
Excessive antropogenic emission, stable meteorological conditions, and regional
transport of air pollutants are important factors in the formation of heavy air pollution
(Zhang et al., 2012a; Guo et al., 2016a; Ning et al., 2018). Under stagnant
meteorological conditions, weak winds, strong and thick temperature inversion layers,
sinking motion and low mixing layer heights are extremely unfavourable for the
diffusion of air pollutants, with these local meteorological conditions acting as an
external driver for the formation of heavy air pollution (Wang et al., 2013a; Xu et al.,
2016; Guo et al., 2016b; Ding et al., 2017).
Regional transport of air pollutants involves complex atmospheric physical and
chemical processes and is influenced by many factors such as meteorological
background field, topography, distribution of emission sources, and atmospheric
chemical transformation (Wang et al., 2013b). The transport and accumulation of air
pollutants result from the interaction between topographic conditions and
meteorological conditions (Zhang et al., 2012b). Air pollutant transport under the
control of the cold front system has a significant impact on the atmospheric level of
$PM_{2.5}$ in Shanghai in winter (Xu et al., 2016). The weather processes leading to heavy
pollution in Hong Kong mainly involve the transport of high-concentration aerosols
carried by cold air (Yang et al., 2019). In the Pearl River Delta, 50-70% of $PM_{2.5}$
originates from long-distance transport, resulting in more severe pollution in the
suburbs than in the nearby urban areas (Wu et al., 2013a). During the aerosol
accumulation stage, $PM_{2.5}$ concentration in the Beijing-Tianjin-Hebei region increased
from 24.2 to 289.8 $\mu g\,m^{-3}$, with the contributions of regional transport increasing
from 12 % to 40 %, while the contribution of local emissions decreased from 59 % to
38 % (Chen et al., 2019). The rapid changes in the formation and dissipation process
of heavy pollution in Beijing are mainly caused by the regional transport and the
alternation between northerly and southerly air masses (Gao et al., 2016), and the
regional transport is estimated to contribute to as much as 53%-70% in the explosive
growth period of $PM_{2.5}$ (Li et al., 2017a). Fine particulates can be transported across a



wide range and over a long distance with obvious trans-boundary transport
characteristics, which has an important effect on the dynamics of atmospheric
pollution (Kim et al., 2012; Khuzestani et al., 2017; Li et al., 2019; Yuan et al., 2019).
Hunan and Hubei provinces in the middle Yangtze River area have a special
geographic location, forming the border of China's most concentrated areas of
atmospheric pollution—the Central Plains, Fenwei Plain, North China Plain, and
Yangtze River Delta region. Hunan and Hubei provinces are located in the downwind
direction of the areas of heavy air pollution sources under the influence of winter
monsoon, serving as a hub for the regional transport of air pollutants (Figure 1). The
mechanism of regional air pollutant transport on the air pollution in the Hunan–Hubei
Plain is a pressing issue of atmospheric environmental science calling for in-depth
investigation. With the development of China's Yangtze River Economic Belt and
middle Yangtze Basin, the atmospheric environment problems in Hunan and Hubei
provinces have become increasingly prominent. Different from cumulative heavy
pollution in central and eastern China, the pollutant-transport characteristics and the
effect mechanism of meteorological conditions on heavy pollution processes in
typical regions like Hunan and Hubei provinces have not yet been systematically
investigated. To fill the knowledge gap, this study employed regional surface
environment data, meteorological network data, atmospheric sounding data, and
ERA-interim reanalysis data in combination with synthetic analysis, climate diagnosis,
and numerical simulation to comprehensively investigate the mechanism of air
pollutant transport on the process of heavy air pollution in winter in Hunan and Hubei
provinces. The findings are of practical value in broadening the scientific
understanding of the formation mechanism differences of heavy atmospheric
pollution between various regions of China and promoting environmental and
ecological protection of the middle Yangtze River area.

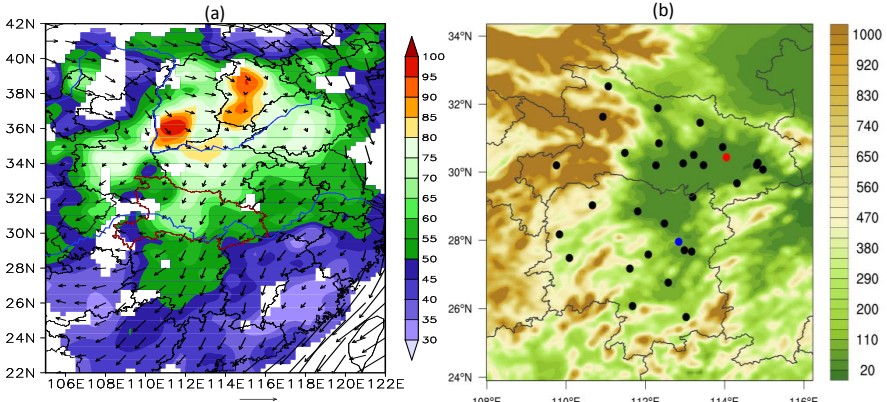


Figure 1 (a) Distribution of average PM$_{2.5}$ concentration (colour scale, in μg m$^{-3}$) and
average ERA-interim 10-m wind vectors (denoted by arrows, in m s$^{-1}$) measured at
the environmental monitoring stations of central and eastern China in January
2015-2019, and (b) topographic map of the distribution of environmental monitoring
stations in 31 cities of Hunan and Hubei provinces (colour scale, in m), where the red
point represents the Wuhan station, and the blue point represents the Changsha
station.

**2 Data and methods**
**2.1 Data source**
The daily average PM$_{2.5}$ concentration in January 2015-2019 was obtained from
China's National Ambient Air Quality Monitoring Network, which can be accessed
through the data centre of the Ministry of Ecology and Environment
(http://datacenter.mee.gov.cn/). For the same period, the hourly data of the surface
meteorological elements such as sea level pressure (SLP), 2-m air temperature, and
10-m wind speed and wind direction, as well as the atmospheric sounding data of
temperature and wind speed at 00 UTC each day at the Wuhan station and Changsha
station, were sourced from China Meteorological Data Network (http://data.cma.cn/).
Moreover, the contemporary ERA-interim daily reanalysis data were obtained
(http://apps.ecmwf.int/datasets/data/) with a resolution of 0.25 °×0.25 ° consisting of
the atmospheric boundary-layer height, SLP, 2-m air temperature, 10-m wind vector
components u and v at 00 UTC each day, as well as the geopotential height,
temperature, vertical speed, and wind vector components u and v at different vertical
layers at 00 UTC each day.






## 2.2 Multivariate and extended empirical orthogonal function decomposition (MV-EOF, EEOF)

The multivariate empirical orthogonal function decomposition (MV-EOF) is essentially the same as extended empirical orthogonal function decomposition (EEOF). They are both based on the classical empirical orthogonal function decomposition (EOF) to extend the spatial dimension of original data matrix in a different manner, that is, MV-EOF allows extension of multiple elements, while EEOF allows extension of multiple consecutive time points.

When using MV-EOF to study the spatio-temporal variation of the transport modes of pollutants, it is necessary to consider the eigenvectors of a combined ensemble of pollutant data and wind pressure field data. By extending the spatial dimension of $PM_{2.5}$ and wind pressure fields, it is possible to derive the distribution of typical multivariate modes and their time coefficients.

The EOF decomposition can reveal the spatial distribution structure of the element field, but the revealed structure is in a time-invariant form, failing to provide a time-dependent spatial distribution structure under disturbance. Based on the significant temporal autocorrelation of element field, EEOF can extend the original data matrix into observation data corresponding to multiple consecutive time points using a certain lag time, so that the time-dependent spatial distribution characteristics and regional transport patterns of pollutants can be obtained. When using EEOF to analyse the quasi-periodic oscillation and evolution characteristics of the region transport of $PM_{2.5}$, it is necessary to first separate the specific periodic components in the element field of $PM_{2.5}$, and then, perform EEOF decomposition on the separated data. The significant periods in the time series of $PM_{2.5}$ can be extracted using a power spectrum method, and the separation of specific periodic components from the element field can be achieved via a second-order Butterworth band-pass filter.

Moreover, other statistical methods such as synthetic analysis and correlation analysis were also applied in this study. It is noteworthy that the anomalies of each element used in the synthesis of a typical example were calculated using the average meteorological values in January 2015–2019.

## 3 Selection of regional air pollutant transport events

There are three main causes of heavy air pollution, namely anthropogeinc

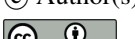



emissions far exceeding the environmental carrying capacity, unfavourable
meteorological conditions, and regional transport of air pollutants. The local
meteorological conditions leading to heavy cumulative pollution mainly include (1)
poor air mobility and calm surface winds or small surface winds (wind speed less than
2 m s$^{-1}$), which are extremely unfavourable for the horizontal diffusion of pollutants;
(2) stable atmospheric conditions, near-surface temperature inversion, and significant
decrease of boundary-layer height, which are unfavourable for the vertical distribution
of air pollutants. In contrast to the above conditions are the meteorological conditions
leading to regional transport of heavy air pollution, which mainly include obviously
enhanced surface winds and favourable horizontal diffusion conditions; under these
transport-facilitating conditions, the pollutant concentration rises, which is mainly
attributed to pollutant transport from the upstream areas, that is, the regional transport
promotes a rapid local accumulation of pollutants.
The atmospheric circulation during the East Asian winter monsoon drives the
regional transport of air pollutants in central and eastern China and facilitates the
accumulation of air pollutants in the middle reaches of the Yangtze River (Yu et al.,
2020). The PM$_{2.5}$ concentration in Wuhan usually rises rapidly under the influence of
strong northerly winds (Lu et al., 2017, 2019b). The regional transport of air
pollutants in the middle Yangtze River region is dominated by near-surface northerly
airflow, which is air pollutant transport from northern China ( Yue et al., 2016; Li et
al., 2019; Xiao et al., 2020). As shown above, the transported air pollutants in Hunan
and Hubei provinces is characterized by high regional concentration of PM$_{2.5}$, north–
south pressure gradient, and anomalous northerly winds. Therefore, MV-EOF was
adopted in this study to extract the common eigenvectors and time coefficients of four
variables—daily average PM$_{2.5}$ concentration, 10-m wind vector components U and V,
and SLP (00 UTC each day) in January 2015–2019 at 31 urban environmental
monitoring stations of Hunan and Hubei provinces.
Figure 2 presents the spatial distribution and time coefficients of the MV-EOF
mode featured by high PM$_{2.5}$ concentration, south–north pressure gradient, and
anomalous northerly winds for the regional pollutant transport in Hunan and Hubei
provinces. In particular, the Wuhan-centred urban agglomeration and Changsha–
Zhuzhou–Xiangtan urban agglomeration had the most obvious spatial characteristics.
This mode accounted for 12% of the total variance, and its time coefficient accounted
for 48.4% of the total variance of the time series of daily averaged PM$_{2.5}$ in the two



provinces, indicating that regional transport was almost the most dominant factor
determining the changes in pollutant concentrations in the Hunan and Hubei
provinces. After normalization of the time coefficient, eight typical regional transport
events with maximum standard deviation of time coefficient were selected to explore
the meteorological conditions leading to regional transport of heavy pollution in the
Hunan and Hubei provinces (Table 1).

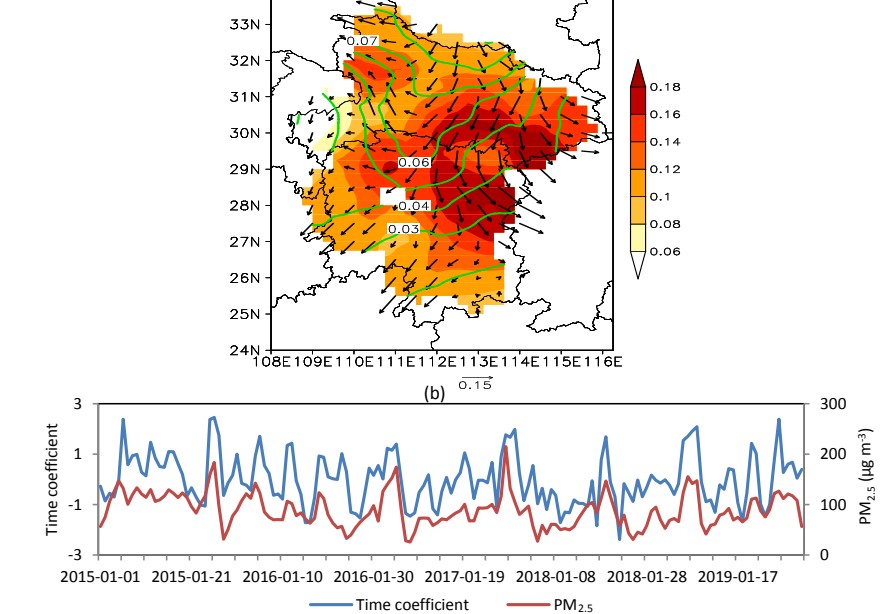



Figure 2 Spatial distribution of the MV-EOF mode (a) of $PM_{2.5}$ denoted by the colour
scale, of wind vector with components U and V represented by arrows, and of SLP
displayed by green contour lines (all the elements are dimensionless), and (b) its time
coefficient and the time series of regional mean $PM_{2.5}$ concentration (in µg m$^{-3}$).

Table 1 Eight typical regional pollutant-transport events in the Hunan and Hubei
provinces

| No. | typical regional transport events (standard deviation of time coefficient) | regional mean $PM_{2.5}$ concentration (anomalies) (µg m$^{-3}$) | regional mean wind speed (anomalies) (m s$^{-1}$) | start and end of regional transport |
|---|---|---|---|---|
| 1 | Jan. 6, 2015 (2.38) | 135 (45) | 2.9 (2.1) | Jan. 4-7, 2015 |
| 2 | Jan. 25, 2015 (2.37) | 161 (71) | 1.6 (0.7) | Jan. 23-27, 2015 |





| 3 | Jan. 26, 2015 (2.46) | 185 (95) | 0.9 (0.1) | Jan. 24-27, 2015 |
| 4 | Jan. 5, 2016 (1.71) | 123 (33) | 2.4 (1.6) | Jan. 3-6, 2016 |
| 5 | Jan. 30, 2017 (1.98) | 104 (14) | 2.5 (1.7) | Jan. 28-31, 2017 |
| 6 | Jan. 7, 2019 (1.91) | 143 (53) | 1.5 (0.7) | Jan. 4-9, 2019 |
| 7 | Jan. 8, 2019 (2.09) | 146 (56) | 1.2 (0.4) | Jan. 5-9, 2019 |
| 8 | Jan. 26, 2019 (2.38) | 125 (34) | 1.7 (0.9) | Jan. 24-27, 2019 |


## 4 Air pollutant transport characteristics in Hunan and Hubei provinces and mechanism of meteorological conditions on the heavy air pollution process

### 4.1 Air pollutant transport and the predictive regional characteristics

Figure 3 presents the spatial distribution of regional $PM_{2.5}$ concentration, anomaly, and surface synoptic situations obtained from eight typical pollutant-transport events on the onset day (0d) and two days earlier (-2d), respectively. As shown in the figure, the region at -2d was under the influence of a uniform pressure field and low air pressure, and a large range of stagnant weather patterns appeared in northern and central China, including an obvious characteristic of small southerly winds and calm winds, an obvious decrease in the atmospheric boundary-layer height, and a weak diffusion of pollutants in the horizontal and vertical directions, all jointly resulting in local cumulative heavy pollution. In particular, the air pollutant concentration was the highest in northern China and the Central Plains, with local $PM_{2.5}$ concentrations exceeding 250 μg m$^{-3}$, whereas the local air pollutant accumulation in Hunan and Hubei provinces occurred at a relatively low level.

On the onset day of regional transport, atmospheric circulation underwent changes, that is, the surface south–north pressure gradient was enhanced. Affected by the lower part of the surface high-pressure system, the eastern cold air, which was dominated by north-easterly winds moved southwards, and the 2-m air temperature was reduced. The wind speeds in the two provinces and their upwind areas were increased obviously, and the atmospheric boundary layer was significantly elevated, with the Hunan–Hubei Plain located at the centre of an anomalous northerly wind belt. On the contrary, the wind speed in southern China was weakened and buffered the northerly airflow, which promoted the horizontal transport and accumulation of near-surface pollutants in the Hunan and Hubei provinces. With respect to pollution evolution over different regions, the $PM_{2.5}$ concentration in northern China and the Central Plains decreased, whereas the $PM_{2.5}$ concentration in the Hunan and Hubei provinces increased.

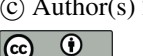



With respect to the observed anomalous wind fields and PM$_{2.5}$ concentration
distribution in Hunan and Hubei provinces (Figure 3e, f), there existed anomalous
southerly winds in the early stage, accompanied by local pollutant accumulation;
when regional air pollutant transport occurred, the wind fields turned into anomalous
northerly winds, which resulted in increased air pollutant concentration in the Hunan–
Hubei Plain, especially in the Wuhan-centred urban agglomeration and the Changsha–
Zhuzhou–Xiangtan urban agglomeration where most areas experienced more than 50
µg m$^{-3}$ of PM$_{2.5}$ concentration anomalies, indicating that regional   transport of air
pollutants would increase the rapid local accumulation of air pollutants in these
high-density cities.

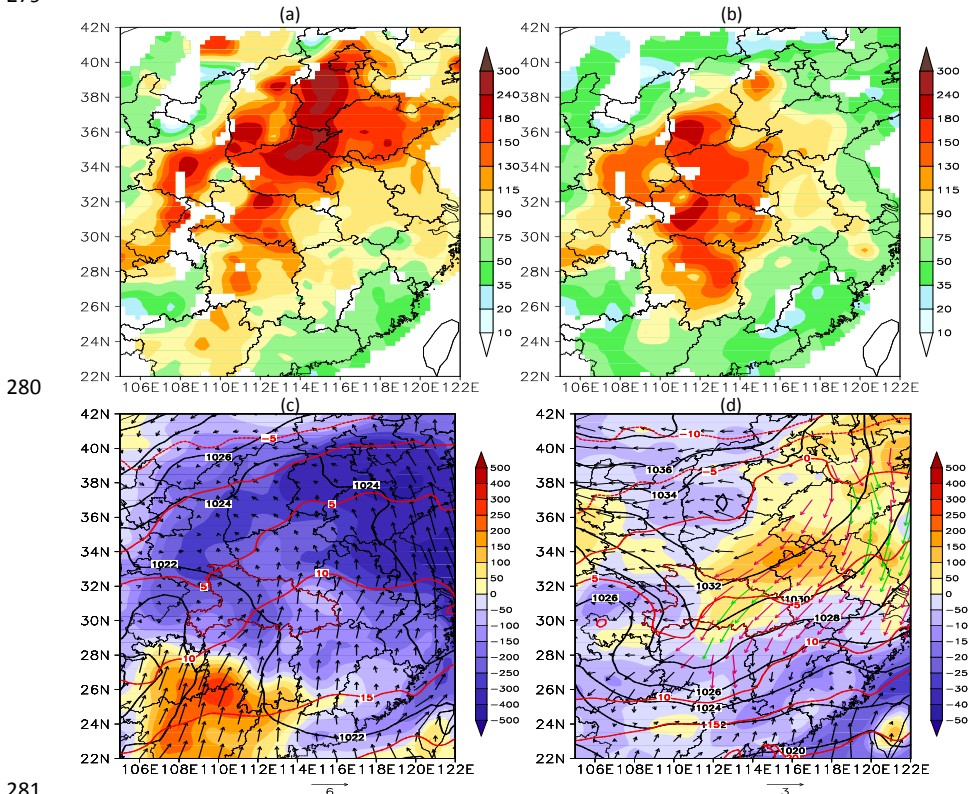



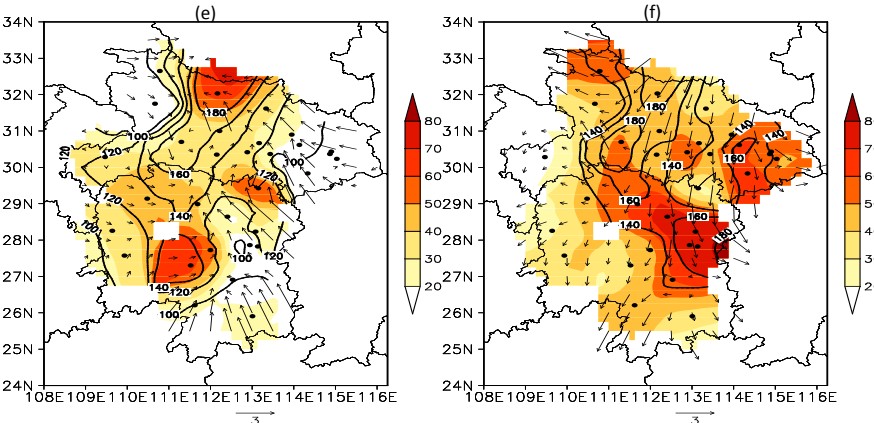


Figure 3 The spatial distribution of daily mean PM$_{2.5}$ concentration (in μg m$^{-3}$) in
central and eastern China on the onset day (b) of eight typical pollution events and
two days earlier (a); weather maps at the surface from the ERA-interim daily dataset
for central and eastern China on the onset day (d) and two days earlier (c), where the
colour scale indicates the anomalies (in m) of atmospheric boundary-layer height, the
black contour lines represent SLP (hPa), the red contour lines represent the 2-m air
temperature (in ℃), the arrows denote anomalous wind fields (in m s$^{-1}$), the red
arrows represent more than 1.5 m s$^{-1}$ of anomalies of wind speed, and the green arrow
represents more than 2.5 m s$^{-1}$ of anomalies of wind speed; spatial distribution of the
daily mean PM$_{2.5}$ concentration (black contour lines, in μg m$^{-3}$) and anomalies (colour
scale, in μg m$^{-3}$) as well as anomalous wind fields (arrows, in m s$^{-1}$) on the above
onset day (f) and two days earlier (e) in the Hunan and Hubei provinces, with data
from the observation stations of the two provinces.

Figure 4 shows the spatial distribution of 10-m wind speed observed at
nationwide observation stations at 00UTC on the onset day of eight typical
pollutant-transport events and the spatial distribution of 24-h temperature changes
observed at the above stations. As shown in the diagrams, the Hunan and Hubei
provinces as well as most of the eastern upstream area experienced a wind speed
above 2 m s$^{-1}$, and the pollutants mainly passed through two transport pathways,
namely the Nanxiang Basin-Yunmeng Plain pathway and the Dabie Mountain's Hilly
Area-Yunmeng Plain pathway, where the local winds moved at a speed of above 3.5
m s$^{-1}$, transporting air pollutants towards the Hunan–Hubei Plain. The dynamic
conditions for regional pollutant transport were triggered by the southward movement

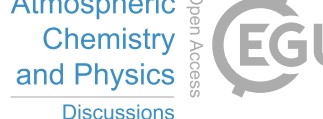

of weak cold air, which was similar to the process of a cold front system causing air
pollutant transport from northern China to the Yangtze River Delta region (Kang et al.,
2019). Under the influence of cold air, most of the northern regions experience
temperatures of -5 to -2 ℃ within 24 h, and the 24-h temperature changes in Hunan
and Hubei provinces were -3 to -2 ℃. Because of the weak southward cold air, its
influence exists as far as only the Hunan province. After passing through the Hunan–
Hubei Plain, the cold air degenerates, and the Guangdong-Guangxi hilly regions and
Zhejiang-Fujian hilly regions in the south undergo positive temperature changes of
0-2 ℃, so wind speed reduces in these areas, causing air pollutants to stagnate and
accumulate in the Hunan–Hubei Plain. As shown above, short-term activities of weak
cold air are crucial for pollutant transport in the Hunan and Hubei provinces.

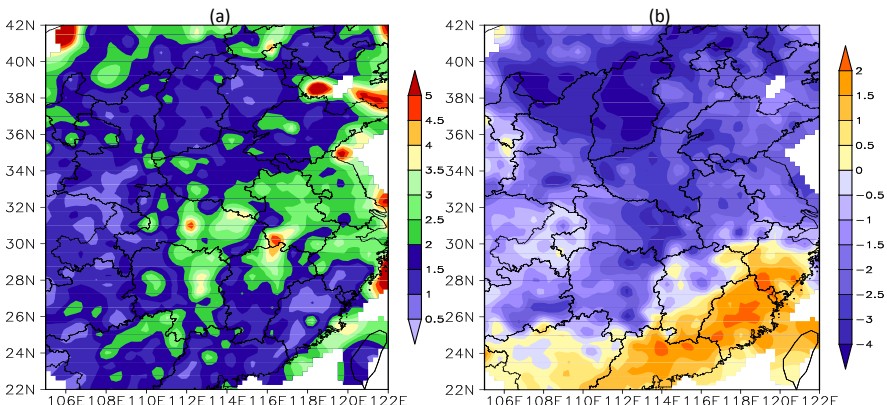

Figure 4 Spatial distribution of surface 10-m wind speeds (in m s$^{-1}$) during eight
typical pollution events (a) and spatial distribution of 24-h temperature changes (in ℃)
during the above events, with the data from nationwide observation stations at 00
UTC.

**4.2 Circulation characteristics affecting regional transport**
Figure 5 represents the spatial distribution of the correlation coefficient between the
time series (sequences in Figure 2b) of a mode of regional pollutant transport and
some meteorological parameters (SLP, boundary-layer height, and 10-m wind vector
with components U and V) to verify the association between regional pollutant
transport and surface synoptic situation. The results indicate that consistent with the
analysis results of Figure 2d, the favourable meteorological conditions for air
pollutant transport in the Hunan and Hubei provinces include anomalous northerly
winds under the influence of the lower part of a surface high-pressure system and a
high atmospheric boundary layer in the transport pathways of the Hunan–Hubei Plain
and in its upstream area, all of which are conductive to diffusion and transport of air
pollutants.

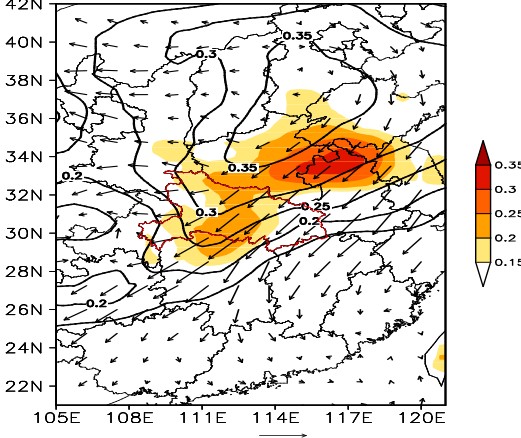


Figure 5 Spatial distributions of the correlation coefficients between the regional
transport mode time coefficient and the meteorological parameters (SLP,
boundary-layer height and 10-m wind vector) during the same period. The correlation
coefficients of SLP denoted by black contour lines, the correlation coefficients of
boundary-layer height denoted by the colour scale, and the correlation coefficients of
10-m wind vector denoted by arrows.

Figure 6 presents the surface synoptic situation and the synoptic situation at
higher layers (850 hPa, 700 hPa, and 500 hPa) caused by eight typical
pollutant-transport events. As shown in the figure, affected by the anomalous
northerly airflow under the influence of the lower part of the surface high-pressure
system in northern China, weak cold air penetrated southward to central China,
driving air pollutants from the northern source areas to Hunan and Hubei provinces.
The southward airflow showed a weaker convergence in southern and eastern China;
moreover, warm air masses converged in the Hunan and Hubei provinces, acting as a
buffer, so air pollutants were stagnated and accumulated in the Hunan–Hubei Plain,
which accelerated the rapid local accumulation of pollutants. The synoptic situation at
850 hPa was similar to that of the surface, that is, the anomalous northerly winds
carried the north-sourced pollutants to Hunan and Hubei provinces, and the



convergence zone with positive anomalies of temperature tended to be uplifted
northward, which made the upper atmospheric layer over the Hunan and Hubei
provinces tend to be in a stable condition. At the mid-high layers of 700 hPa and 500
hPa, Hunan and Hubei provinces and their upstream area were markedly characterized
by the presence of a "warm lid" under the influence of positive geopotential height
anomalies, which prevented pollutants from diffusing to higher atmospheric layers
during the transport process; moreover, a warm high-pressure system existed in the
mid-high layers. All these characteristics indicated that it was a weak cold air system
with a shallow boundary layer.

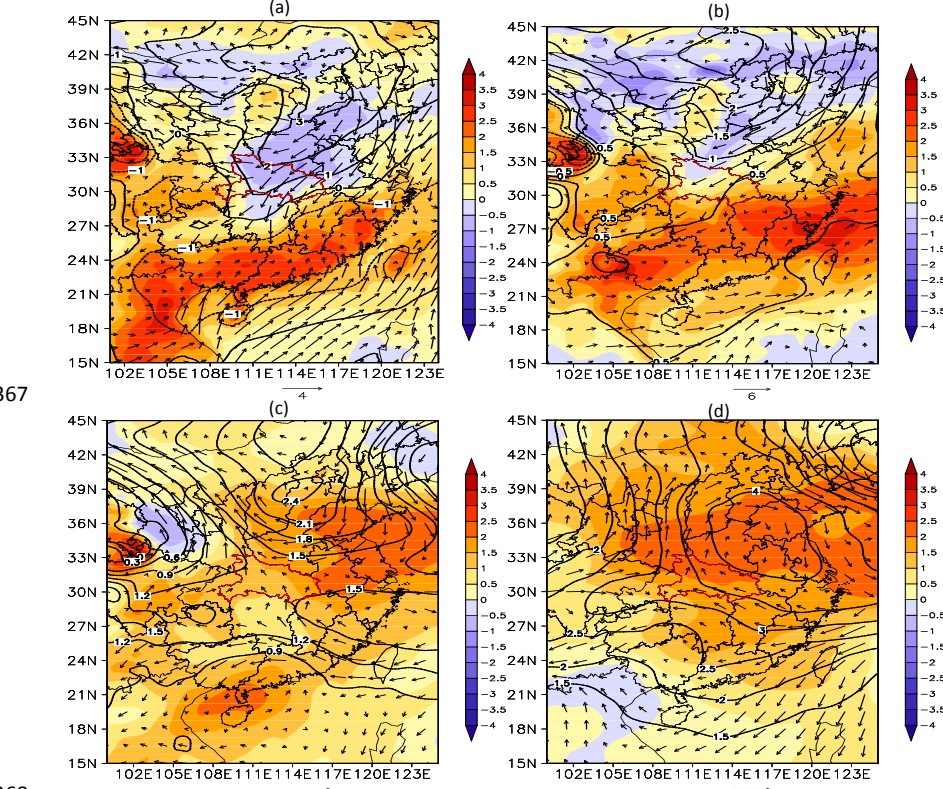



Figure 6 The surface synoptic situation (a) and the synoptic situation at higher layers
of 850 hPa (b), 700 hPa (c), and 500 hPa (d) synthesised from eight typical
pollutant-transport events. The black contour lines represent the anomalies of surface
SLP (in hPa) and high-altitude geopotential height (in dagpm); the colour scale
represents temperature anomalies (in ℃); the arrows represent anomalous wind fields
(in m s$^{-1}$) from the ERA-interim daily dataset at 00:00 UTC.






The atmospheric circulation situation on pollution-free days showed that strong
cold-air advection with strong winds would have an obvious positive effect on the
removal of air pollutants. To better understand the differences between the
meteorological conditions for air pollutant transport into the Hunan and Hubei
provinces and the meteorological conditions for air pollutant removal from the region,
the synoptic situation for strong winds to remove pollution was synthesized from
eight events in which air pollutants were removed by strong winds. Next, the
element-wise differences of the synoptic situation between the above two types of
events were calculated, as shown in Figure 7. For the surface synoptic situation, there
was no significant difference in the 10-m wind vectors between the two types of
events in the Hunan and Hubei provinces, but in the case of strong winds removing
the pollutants, the cold high-pressure system in northern China was stronger, with
strong cold airflow sweeping across the Hunan and Hubei provinces and directly
affecting the coastal areas of southern and south-eastern China. The wind speeds in
the coastal areas were anomalously high, where no warm air masses converged to
buffer pollutant transport, so air pollutants failed to accumulate in the Hunan and
Hubei provinces. Moreover, the differences in the synoptic situation at the mid-high
layers indicated that when strong winds were removing the air pollutants, the
northerly winds were stronger, the cold air masses were thicker, and no stationary
atmospheric states could be formed in the mid-high layers above Hunan and Hubei
provinces, which facilitated vertical diffusion of pollutants.

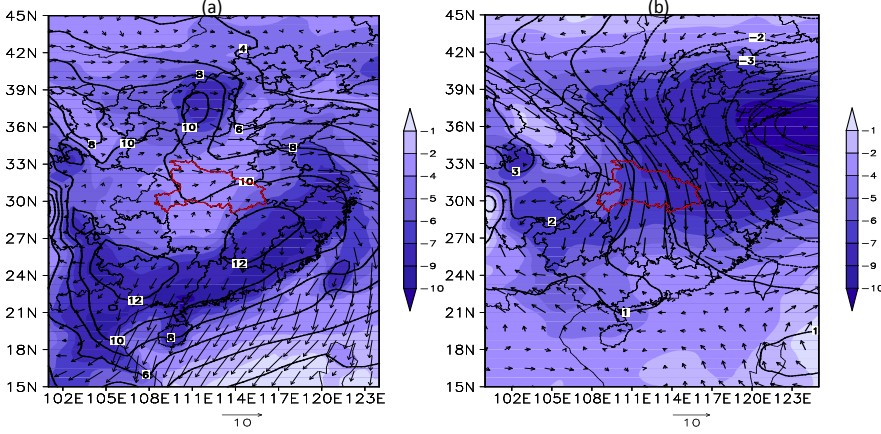


Figure 7 Element-wise differences in synthesized synoptic situations between eight



events of big wind-induced pollutant removal and eight events of big wind-induced
pollutant input at the surface (a) and the atmospheric layer of 700 hPa (b), with the
black contour lines representing the surface SLP differences (in hPa) and the
high-altitude geopotential height differences (in dagpm), the colour scale representing
the temperature differences (in ℃), and the arrows representing the wind field
differences (in m s$^{-1}$) from the ERA-interim daily dataset at 00:00 UTC.

**4.3 Influence of atmospheric vertical structure on regional transport**
Figure 8 presents the longitudinal cross-sectional structure of the pollutant-transport
channel along the Hunan–Hubei Plain (synthesised from eight typical
pollutant-transport events). There was a clear longitudinal vertical circulation above
the plain, which spanned the latitudes of 26.5 ℃N–32.5 ℃N and vertically extended to
the layer of 650 hPa, with the central height corresponding to an air pressure of 800
hPa. The mid-high layers above the 800-hPa layer were dominated by weak southerly
airflow, whereas below the 800-hPa layer, strong northerly airflow was observed, with
wind speed anomalies reaching 2–5 m s$^{-1}$. The northerly airflow rose in the south of
the plain, but it underwent circulation and sank in the north of the plain. The
longitudinal vertical circulation structure was triggered by the southward penetration
of the weak cold air with a shallow boundary layer. The southward moving weak cold
air was wedged into the bottom of the warm air, and the warm air was uplifted, not
only leading to stable stratification similar to that in the frontal zone but also forming
a high-altitude "warm lid" above the Hunan–Hubei Plain and the upstream area,
which suppressed the vertical diffusion of pollutants during their transport. The cold
air exhibited weak activity, and wind speed anomalies turned negative in the
downstream area, where warm air was squeezed and underwent convergence, rising
and moving northward in a roundabout manner. Later, the warm air transformed into a
sinking airflow when blocked by the high-altitude "warm lid", which resulted in air
pollutant accumulation and completion of the longitudinal vertical circulation.
Vertical circulation played an important role in the transport and formation of heavy
pollution in the Hunan and Hubei provinces, promoting the north-sourced pollutants
to migrate and accumulate towards the Hunan–Hubei Plain. In addition, the stable
stratification at mid-high layers inhibited the escape of air pollutants, thereby
confining them within the atmospheric boundary layer and promoting rapid
accumulation of air pollutants in the plain.

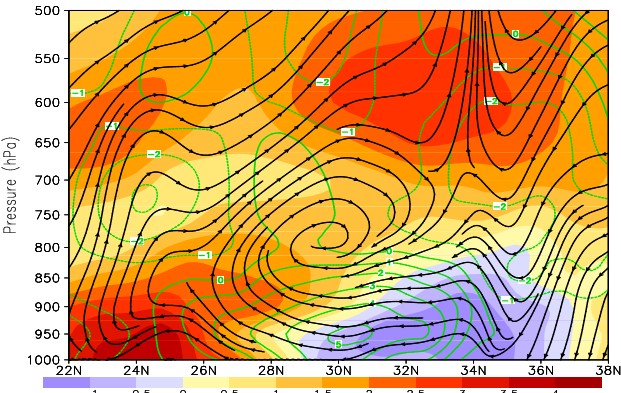


Figure 8 The longitudinal cross-sectional distribution of meteorological conditions
synthesised from eight typical pollutant-transport events (112.25 °E–113 °E on
average), where the black flow lines represent longitudinal circulation (synthesised by
*v* and *w,* vertical velocity is multiplied by 10), the green contour lines represent wind
speed anomalies (in m s$^{-1}$), and the colour scale represents temperature anomalies
(in °C).

Based on the daily atmospheric sounding data at 00UTC recorded at the Wuhan
station and Changsha station in Hunan and Hubei provinces, vertical sounding
profiles of temperature and wind speed were synthesised from eight typical
pollutant-transport events (Figure 9). It is evident that during pollutant transport, the
surface wind speeds at the Wuhan station reached 5 m s$^{-1}$; in near-surface layers
below 950-hPa, wind speeds and air temperature increased and decreased with the
layer height, respectively, indicative of the non-stagnant state of the near-surface
atmospheric structure and the "openness" of the transport channel, which facilitated
the transport of a large amount of pollutants to the Hunan and Hubei provinces. In
contrast, there appeared an obvious temperature inversion layer at 950-900 hPa,
where the wind speeds decreased with height, forming stable atmospheric
stratification which suppressed the vertical diffusion of pollutants during their
transport; at 800-750 hPa, there existed another stable stratified structure, which
confined air pollutant transport within the atmospheric boundary layer. Analysis of the
sounding data from the Changsha station gave a consistent result, but the transport
channel gradually "shrank". The atmosphere below 975 hPa was in a non-stationary
state. Above 975 hPa, the temperature inversion layer was thin, but the isothermal
layer was thicker.

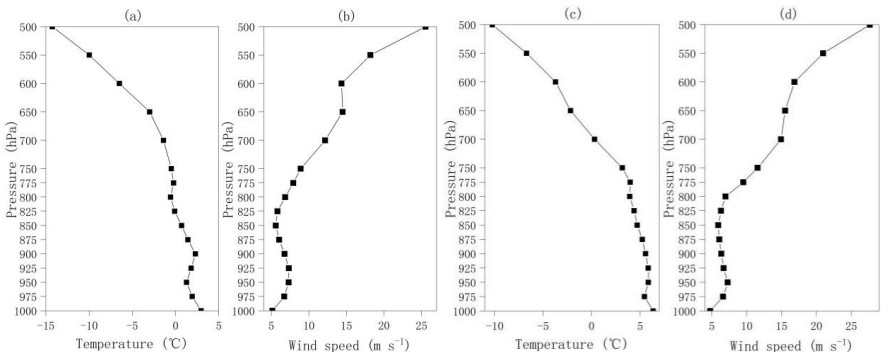

Figure 9 Atmospheric sounding profiles of temperature (a, c) and wind speed (b, d) of
the Wuhan station (a, b) and Changsha station (c, d) synthesised from eight typical air
pollutant transport events
Regional means of relevant meteorological parameters were calculated separately
for the upstream area of Hubei province, Hubei province, and Hunan province and the
downstream area of Hunan province to further investigate the vertical variation of
meteorological conditions during pollutant transport in different areas (Figure 10).
The results showed that in the Hunan and Hubei provinces and their upstream areas,
(1) the temperature anomalies below 950 hPa decreased with height, which was
consistent with the atmospheric sounding results of the meteorological stations; (2)
below 850 hPa, there existed anomalous northerly winds with high speeds, with the
atmosphere exhibiting non-stationary characteristics; and (3) at mid-high layers, the
temperature anomalies significantly increased with height, where there existed
anomalous southerly low-speed winds and stable atmospheric stratification, with
anomalous "warm lid" characteristics. The low-altitude wind fields diverged in the
Hubei province and its upstream area but converged in the Hunan province and its
downstream area, leading to air pollutant transport and accumulation. In the
downstream area, the anomalous winds turned south, buffering transport of air
pollutants.

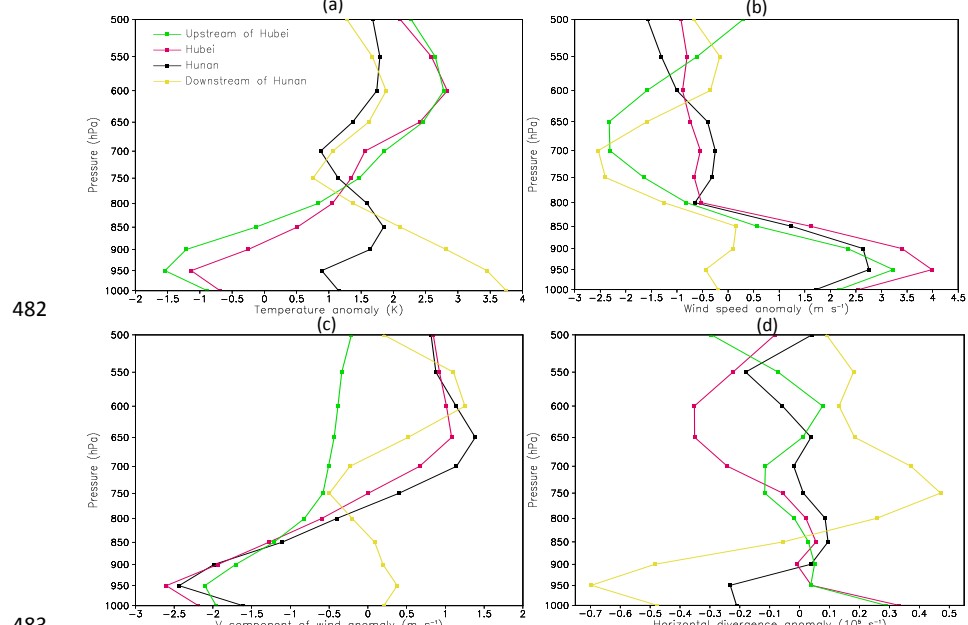



Figure 10 Synthesized vertical profiles of temperature anomaly (a), wind speed anomaly (b), v component of wind anomaly (c), and horizontal divergence anomaly (d) in different geographical areas during eight typical pollutant-transport events, where the green lines represent the upstream area (33-36 °N, 112-118 °E), the red line Hubei province (30-33 °N, 111-116 °E), the black line Hunan province (26-30 °N, 110-114 °E), and the yellow line the downstream area (23-26 °N, 109-115 °E).

As shown above, the mechanisms of meteorological conditions on air pollutant transport and the heavy pollution process in the Hunan and Hubei provinces are the following: (1) the anomalous northerly winds in the near-surface layer of the Hunan and Hubei provinces and their upstream area are accompanied by the southward penetration of a shallow cold layer, which provides dynamic conditions for regional pollution transport; (2) weak cold air degenerates as it passes through the Hunan–Hubei Plain, causing warm air to accumulate in the near-surface layer of the downstream area, where winds slow down and converge, buffering pollutant transport and thereby resulting in pollutant accumulation; (3) the atmosphere in the near-surface layer of the Hunan and Hubei provinces is in a non-stationary state (dry intrusion of cold air, anomalously strong winds, and positive anomalies of boundary-layer height), which is conducive to the horizontal transport of pollutants; (4) the mid-high layers



have stable stratification, characterized by temperature inversion and the presence of a
"warm lid", which inhibits the diffusion of pollutants to the upper layers; and (5) there
is obvious longitudinal vertical circulation above the Hunan–Hubei Plain, and the
regional transport is confined within the atmospheric boundary layer, which results in
sinking and accumulation of the pollutants, thereby promoting rapid accumulation of
pollutants in the Hunan and Hubei provinces.

**5 Quasi-4-d periodic oscillation of pollutant transport in the Hunan and Hubei**
**provinces**
**5.1 Quasi-4-d evolution characteristics of regional transport of PM$_{2.5}$**
Regional transport is associated with the synoptic scale. The East Asian winter
monsoon has a short-term 4-6-day cycle of weak cold air activity. Power spectrum
analysis revealed that the time coefficient of the regional transport mode had a
significant period of 5-6 days, which was associated with the short-term weak cold air
activity of the East Asian winter monsoon. To explore the spatial evolution
characteristics of the eigenvector of regional pollution transport, EEOF was used to
analyse the spatio-temporal field of PM$_{2.5}$ and 10-m wind vector in central China
(Hunan and Hubei provinces and the upstream area of the Henan province).
To reveal the effect of the short-term cycle of weak cold air activity on regional
transport, a second-order Butterworth band-pass filter with a 4-6-day filtering window
was applied on a synoptic scale to the time series of PM$_{2.5}$ and 10-m wind vector
collected at 48 urban environmental monitoring stations in central China in January
2015–2019. Next, the time series of each station with a period of 4-6 days was
synthesized to form a data matrix, which was subjected to EEOF decomposition based
on a lag time of 2 days. The first two eigenvectors passed the significance test. The
first and second eigenvectors of PM$_{2.5}$ (10-m wind vector), referred to as EEOF1 and
EEOF2 respectively, cumulatively accounted for 48.9% (37.4%) of the total variance.
The eigenvectors described by EEOF1 and EEOF2 represent situations at different
instances of time during the periodic process, with EEOF1 leading EEOF2 in phase
by a quarter of a period (1 days).
The time coefficient of EEOF eigenvectors for PM$_{2.5}$ had a good correlation with
that for the 10-m wind vector (Figure 11). The EEOF2 time series of PM$_{2.5}$ was
significantly positively correlated with the EEOF1 time series of 10-m wind vector
(coor = 0.5), indicative of the driving of regional transport by strong winds; the





EEOF1 time series of $PM_{2.5}$ was significantly negatively correlated with the EEOF2
time series of 10-m wind vector (coor = -0.6), indicative of the removal of transported
pollutants by strong winds.

540       Figure 11 illustrates the eigenvectors (i.e., spatial distribution of the eigenvectors

EEOF2 and EEOF1 ) of $PM_{2.5}$ in the order of increasing time lags, namely 0 day, 1
day, and 2 days for EEOF2, followed by 0 day for EEOF1. The results showed that
the eigenvector of EEOF1 at a 1-day (2-day) (omitted from illustration) time lag was
similar to that of EEOF2 at a 0-day (1-day) time lag, that is, the four eigenvectors
constituted a 4-day oscillation period of the regional transport. Because of the
correlation of EEOF time coefficient between $PM_{2.5}$ and 10-m wind vector, the
eigenvectors of 10-m wind vector were arranged in the order of EEOF1 (sequentially
at time lags of 0, 1, and 2 days) and EEOF2 (at a time lag of 0 d) to correspond to the
above eigenvectors of $PM_{2.5}$.

550       As shown in Figure 11, the EEOF eigenvectors at different time lags reflected the

spatial evolution characteristics of regional air pollutant transport, and the regional
transport had a quasi-4-d oscillation period, with the synoptic situation undergoing the
same evolution as revealed earlier in this study. On the first day, stable meteorological
conditions occurred and weak winds prevailed in the whole region, with air pollutant
accumulation in the upstream area of northern Henan province. On the 2nd day, the
cold air penetrated southward, and the wind speeds in the upstream area increased,
continuously and cumulatively transporting air pollutants to the Hunan and Hubei
provinces, mainly through the Nanxiang Basin and the low hills of the Dabie
Mountain. On the third day, the northerly winds in the Hunan and Hubei provinces
and their upstream area reached maximum speeds, and the near-surface layer was in a
non-stationary state; the air pollutants were transported along two pathways (the
Nanxiang Basin-Yunmeng Plain pathway and the Dabie Mountain's Hilly
Area-Yunmeng Plain pathway) to the Hunan–Hubei Plain, where they accumulated;
Meanwhile, air pollutants in the upstream area of northern Henan province were
removed by strong winds. The mechanism of air pollutant transport and heavy
pollution formation in the Hunan and Hubei provinces has been discussed earlier in
this study. On the fourth day, strong winds continued to influence the upstream area,
where the local pollutants were fully removed, leading to completion of regional
pollutant transport.


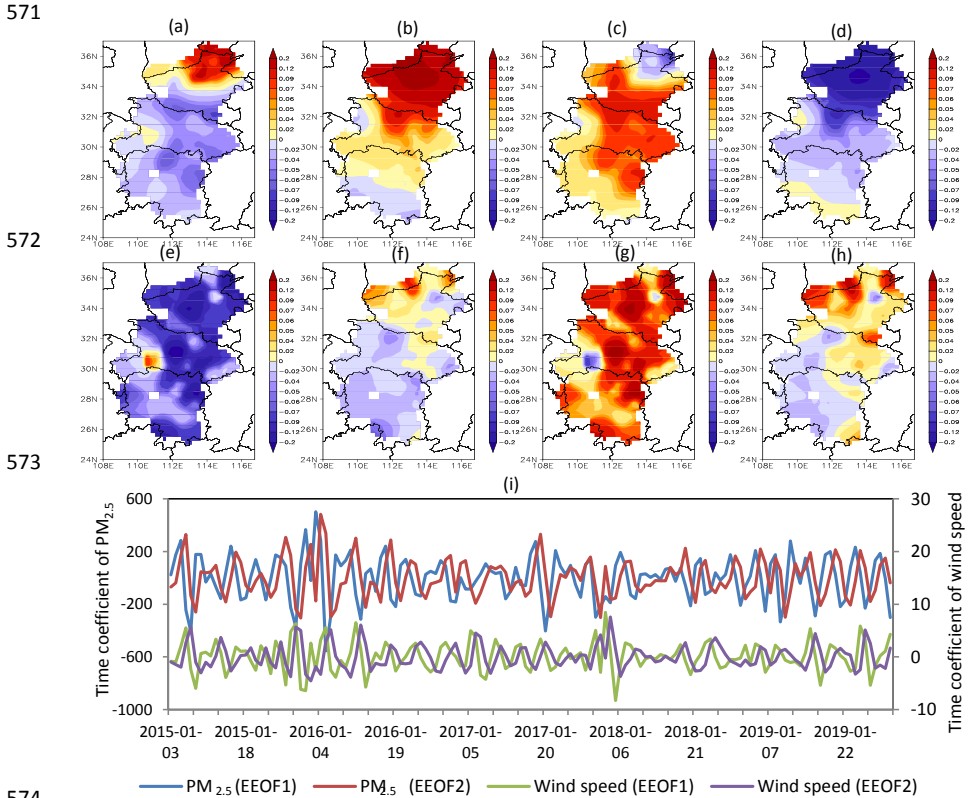




Figure 11 Eigenvectors (a-h) of the EEOF1 and EEOF2 at different time lags for $PM_{2.5}$

and 10-m wind vector during regional pollutant transport and the time coefficients (i).

For $PM_{2.5}$, EEOF2 exhibited time lags of 0 (a), 1 (b), and 2 days (c), and EEOF1

exhibited a time lag of 0 day (d); for the 10-m wind vector, EEOF1 exhibited time lags

of 0 d (e), 1 d (f), and 2d (g), and EEOF2 exhibited a time lag of 0 day (h).

580

**5.2 Numerical simulation validation**

The regional transport event during January 3 and 6, 2016 selected from table 1,

which corresponded to the maximum positive values of the EEOF time coefficient,

was used WRF-Chem model to simulate pollutant-transport characteristics and

meteorological transport conditions in Hunan and Hubei provinces, and a closure

experiment was performed on pollutant emission sources to verify the important

contribution of regional transport to the rapid accumulation of air pollutants in the

Hunan–Hubei Plain.

WRF-Chem is a new-generation mesoscale three-dimensional air quality model





(Grell et al., 2005), which is jointly developed by NOAA, NCAR, and UCAR and
allows on-line coupling of meteorological conditions and atmospheric chemistry. A
central China-specific environmental meteorology numerical model system with the
WRF-Chem model as the core component demonstrated good performance in forecast
evaluation and application (Bai et al., 2016, 2020), and the scheme for localized
testing of the model is documented in detail elsewhere (Bai et al., 2016). The air
pollutant transport process during January 3-6, 2016, in the Hunan and Hubei
provinces was simulated here by the above numerical model system based on the
MIX anthropogenic emissions inventory for Asia in January 2016 (Li et al., 2017b),
with the initial and boundary atmospheric conditions set using the NCAR FNL data
with a resolution of $1° \times 1°$.
Figure 12 shows the evolving spatial distribution of the regional transport flux of
$PM_{2.5}$ at 00 UTC during January 3-6, 2016. The results indicate that the simulation
results were very similar to the EEOF analysis results. The regional pollution
transport was subject to a quasi-4-d period consisting of the following sequential
events: pollutant accumulation in the upstream calm wind zone, pollutant inputs along
the Nanxiang Basin and the hilly area of the Dabie Mountain, pollutant transport and
accumulation in the Hunan–Hubei Plain, and dissipation and removal of regional
pollution.

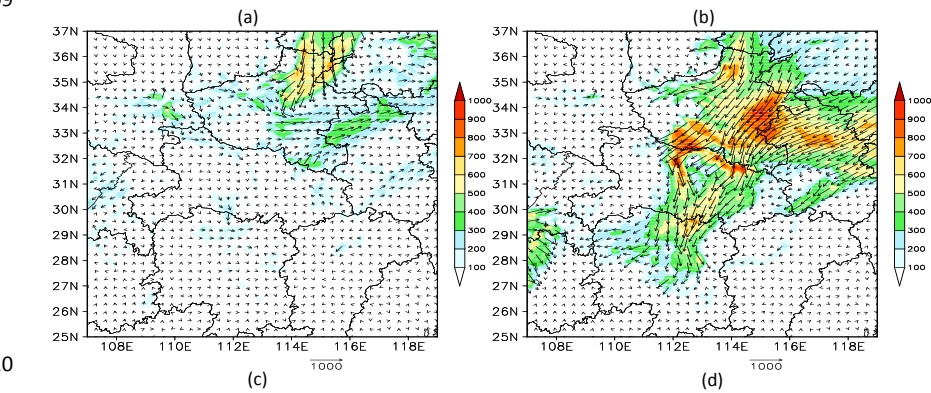






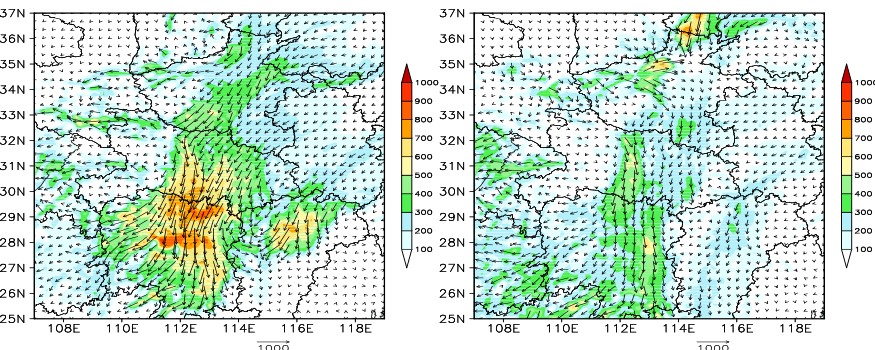

Figure 12 Spatial distribution of the regional transport flux of PM$_{2.5}$ simulated by the WRF-Chem model (the arrows denote PM$_{2.5}$ transport flux, and the colour scales denote modulus length in μg m$^{-2}$ S$^{-1}$) at 00 UTC on January 3 (a), January 4 (b), January 5 (c), and January 6 (d) in 2016.

Figure 13 presents a longitudinal cross-sectional profile of the regional transport pathway within the atmospheric boundary layer along the Hunan–Hubei Plain at different transport times. The results indicated that the main transport pathway of air pollutants was below 1-km height in the atmospheric boundary layer. Air pollutants were inputted and accumulated along the Nanxiang Basin and then transported to the Hunan–Hubei Plain under the driving force of the northerly airflow, whereas the southerly airflow acted as a buffer to prevent air pollutants from moving further southward, which resulted in rapid accumulation of the transported pollutants; the maximum values of PM$_{2.5}$ transport flux were observed near a height of 400-600 m. The atmosphere below 1-km height in the atmospheric boundary layer of the Huan–Hubei Plain was in a non-stationary state, while there existed obvious isothermal stratification in the atmosphere above the pathway, which limited the upward diffusion of pollutants. The simulation results verified the above-mentioned effect mechanism of meteorological conditions on air pollutant transport and heavy pollution in the Hunan and Hubei provinces.



632

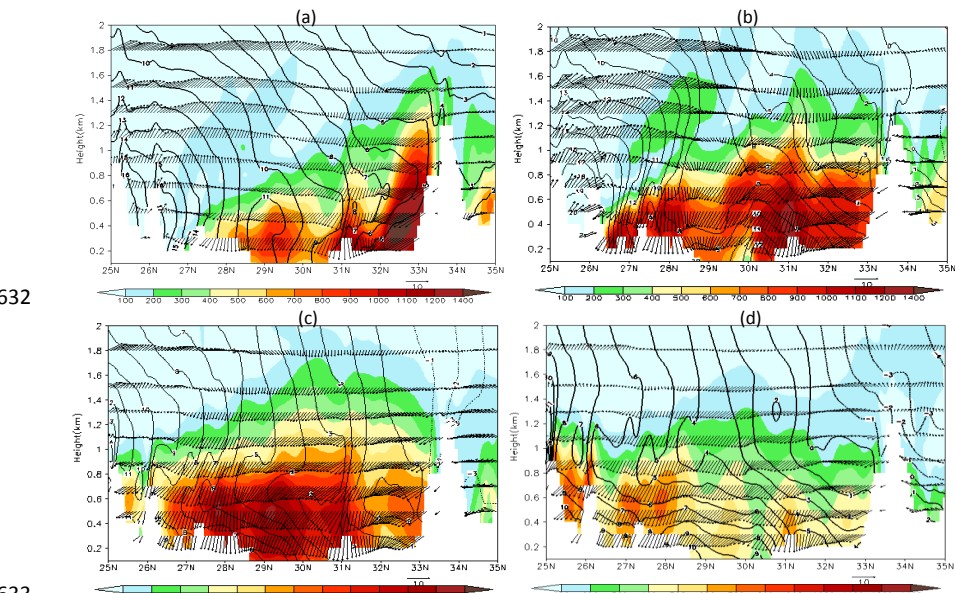

633

Figure 13 Longitudinal cross-sectional profiles simulated by the WRF-Chem model
for pollutant transport along 112.25 °E at 00 UTC of January 4 (a), 12 UTC of January
4 (b), 00 UTC of January 5 (c), and 12 UTC of January 5 (d) in 2016, where the
colour scales represent PM$_{2.5}$ transport flux (in μg m$^{-2}$ s$^{-1}$), the arrows represent
horizontal wind fields (in m s$^{-1}$), and the black contour lines represent temperature
(in °C).

   A closure experiment was conducted by hypothetically shutting down the
anthropogenic emission sources in the Hunan and Hubei provinces and simulating the
PM$_{2.5}$ flux component and flux percentage in the two provinces contributed by PM$_{2.5}$
transport from exogenous sources. Figure 14 presents a simulated time-dependent
vertical profile on the Jianghan Plain in the closure experiment. The results showed
that the vertical distribution of pollutants during their transport was affected by the
northerly winds in the boundary layer. An increase in wind speed led to an increase in
PM$_{2.5}$ transport, with transport from exogenous sources contributing as high as 60-80%
to the near-surface PM$_{2.5}$ and as high as 80-90% to PM$_{2.5}$ at a height of 600-900 m,
which indicated that the regional transport had a significant promotional effect on
pollutant accumulation in the Hunan and Hubei provinces. Because of the instability
of near-surface atmospheric layer, the transported exogenous PM$_{2.5}$ accumulated first
on the surface (18-19 UTC on January 4) and then accumulated continuously for 3-4 h



at the bottom of the isothermal layer at a height near 700 m, which indicated that the
high-altitude accumulation of transported pollutants was closely related to stable
atmospheric stratification.

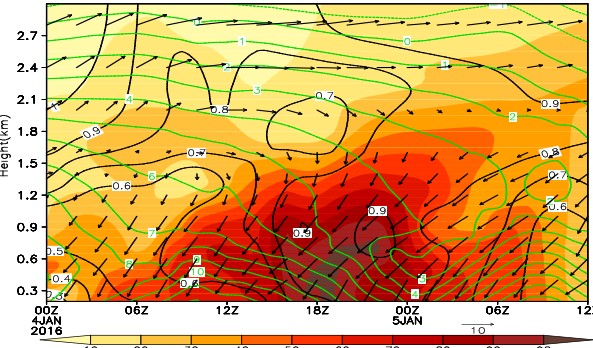

Figure 14 A height-time profile (30 °N, 112.25 °E) simulated by the WRF-Chem model
for pollutant transport after hypothetical closure of the local anthropogenic emission
sources of Hunan and Hubei provinces, where the colour scales represent the
concentration of $PM_{2.5}$ transported from exogenous sources (in μg m$^{-3}$), the arrows
represent horizontal wind fields (in m s$^{-1}$), the green contour lines represent
temperature (in °C), and the black contour lines represent the percent contribution of
exogenous sources to $PM_{2.5}$ transport ($PM_{2.5}$ concentration in the closure experiment
divided by $PM_{2.5}$ concentration in a control experiment).

**6 Conclusions and discussions**

The Hunan–Hubei Plain in the middle Yangtze Basin has a special geographical
location. It forms a border of China's areas with the most concentrated atmospheric
pollution—the Central Plains, Fenwei Plain, North China Plain, and Yangtze River
Delta region—and is located in the downwind direction of the areas of heavy
pollution sources under the influence of winter monsoon, serving as a hub for the
regional transport of atmospheric pollutants in China. The extent and mechanism of
the effect of regional pollutant transport on air pollution in the Hunan–Hubei Plain is
a pressing issue in atmospheric environmental science, calling for in-depth
investigation. Different from cumulative heavy pollution in central and eastern China,
the pollutant transport characteristics and the effect mechanism of meteorological
conditions on heavy pollution processes in typical regions like Hunan and Hubei
provinces have not been systematically investigated yet.

Different from the cumulative heavy pollution under unfavourable local
meteorological conditions, the atmospherically transported heavy air pollution is
under favourable meteorological conditions for horizontal diffusion, that is, the
atmospheric boundary layer is high, the wind speeds in the near-surface layer are
anomalously high, and the $PM_{2.5}$ concentration tends to grow rapidly under the
influence of strong northerly winds. The MV-EOF method was employed in this study
to extract a typical mode of regional pollutant transport, which was characterized by
high $PM_{2.5}$ concentration, south–north pressure gradient, and anomalous northerly
winds in the Hunan and Hubei provinces in January 2015–2019. The time coefficient
of the MV-EOF mode accounted for 48.4% of the total variance of the average $PM_{2.5}$
concentration in the two provinces, which indicated that regional pollutant transport
was almost the dominant factor determining the changes in pollutant concentration in
the region; regional transport caused the $PM_{2.5}$ concentration anomalies in the
Wuhan-centred urban agglomeration and the Changsha–Zhuzhou–Xiangtan urban
agglomeration to exceed 50 μg m$^{-3}$, promoting rapid local accumulation of pollutants
in these high-density cities.
By synthesizing eight typical pollutant-transport events in Hunan and Hubei
provinces in conjunction with surface meteorological observation data, atmospheric
sounding data, and reanalysis data, this study comprehensively analyzed the effect
mechanism of meteorological conditions on air pollutant transport and accumulation
(Figure 15). Transporting air pollutants mainly passed through the Nanxiang Basin
and the low hills of the Dabie Mountain. The anomalous northerly winds in the
near-surface layer of the Hunan and Hubei provinces and their upstream area were
accompanied by southward penetration of a shallow cold layer, which provided
dynamic conditions for regional air pollutant transport. The weak cold air degenerates
as it passes through the Hunan–Hubei Plain, causing warm air to accumulate in the
near-surface layer of the downstream area, where winds slow down and converge,
buffering pollutant transport and resulting in pollutant accumulation; the atmosphere
in the near-surface layer of the Hunan and Hubei provinces is in a non-stationary state
(dry intrusion of cold air, anomalously strong winds, and positive anomalies of
boundary-layer height), which is conducive to the horizontal transport of pollutants.
The mid-high layers have stable stratification, characterized by temperature inversion
and the presence of a "warm lid", which inhibits the diffusion of air pollutants to the
upper layers; there is obvious longitudinal vertical circulation above the Hunan–Hubei





Plain, which confines the regional transport within the atmospheric boundary layer
where the pollutants sink and accumulate, thereby promoting rapid accumulation of
pollutants in Hunan and Hubei provinces. These results are also verified by numerical
simulations.

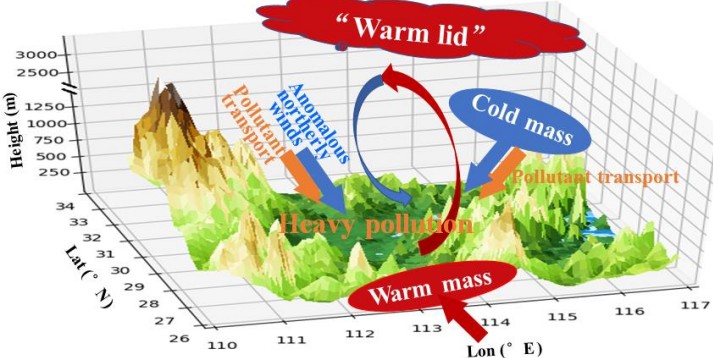


Figure 15. Meteorological mechanism of regional transport in winter heavy air
pollution events in the middle Yangtze Basin areas over China

The activities of weak cold air are crucial for pollutant transport in the Hunan
and Hubei provinces. EEOF analysis revealed that the regional pollutant transport in
the two provinces is subject to a quasi-4-d period consisting of the following
sequential events: pollutant accumulation in the upstream calm wind zone, air
pollutant input along the Nanxiang Basin and the hilly area of the Dabie Mountain,
pollutants transport and accumulation in the Hunan–Hubei Plain, dissipation and
removal of regional pollution; this is associated with the short-term 4-6-day cycle of
weak cold air activity during the East Asian winter monsoon. It is noteworthy that the
continuous accumulation of air pollutants in the upstream area and the sudden
changes of synoptic situation (i.e., a change from a uniform surface pressure field to a
surface high-pressure system whose lower part strongly influences the surface, an
increase of the south–north pressure gradient, and the formation of anomalous
northerly airflow) can be used as a predictive warning sign for air pollutant transport
in the Hunan and Hubei provinces.
In addition, from a geographic aspect, the Hunan and Hubei provinces are in a
large exoreic basin, which is bounded by the Jing Mountain, Dahong Mountain, and
Dabie Mountain to the north, the Wuling Mountains to the west, the Mufu Mountain
and Luoxiao Mountain to the east, and the Xuefeng Mountain and its foothills to the



south and has an area of more than 130,000 km$^2$. It is necessary to have deeper insights into the concentrating effect of this special sub-basin topography and the underlying surface of the complicated basin on regional air pollutant transport, as well as the mechanism of the multi-scale synergistic effect of local circulation, synoptic situation, and East Asian monsoon climate change on regional air pollutant transport.

*Data availability*. The data used in this paper can be provided by Yongqing Bai (2007byq@163.com) upon request.

*Author contributions*. YB , TZ, and YZ conducted the study design. JX, WH, and SK provided the observational data. WH, YG, HZ and LL assisted with data processing. YB wrote the paper with the help of LL and SK. TZ, YZ, JX, LL, and SK were involved in the scientific interpretation and discussion. All authors provided commentary on the paper.

*Competing interests*. The authors declare that they have no conflict of interest.

*Acknowledgment.* The authors are grateful to Associate Professor Zhao Shuyun, Department of Atmospheric Sciences, School of Environmental Sciences, China University of Geosciences (Wuhan) for the useful comments on improving the first draft of this paper.

*Financial support.* This research has been supported by the National Natural Science Foundation of China (grant no. 41830965; 42075186), the National Key Research and Development Project of China (grant no. 2016YFC0203304).

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
