# Peer review of "Yue Zhou1\* Jie Xiong1 Weiyang Hu2 Yao Gu2 Lin Liu1 4 Yongqing Bai1 Tianliang Zhao2\* Shaofei Kong3 Huang Zheng3 5 6 7 1. Hubei Key Laboratory for Heavy Rain Monitoring and Warning Research, Institut"

_Atmospheric Chemistry and Physics, 2020_

## Referee Comment (RC1) · Anonymous Referee #1 · 5 Feb 2021

Review of the paper

Meteorological mechanism of regional transport in winter heavy air pollution events in the middle Yangtze Basin areas over China

by Yongqing Bai and co-authors

The paper describes the meteorological processes that can lead to largely increased air pollution in the Hunan and Hubei provinces in central China in winter. The authors analyse PM2.5 observations together with ERA interim meteorological re-analysis data by means of multivariate EOF decomposition.

[Figure]

The paper suffers from several significant shortcomings. I do not recommend its publication in ACP.

First of all the analysis deals with a very regional or local effect. The authors do not make clear why it should be of interest for an international scientific audience to get insights into the meteorological effects leading to increased pollutants concentrations in the Hunan and Hubei provinces. In particular, the authors do not put their findings in the context of similar studies or similar conditions elsewhere in the world. Also, the literature cited in the paper consists almost exclusively of Chinese authors reporting about air quality in China. Of course, this is necessary in order to put the study into the context of other similar studies. However, the authors should have taken other international studies into account.

Second, the paper repeats its findings several times, partly even by using exactly the same expressions. The explanations given in section 4.2 are more or less repeated in 4.3 (lines 488-501), 5.1 (lines 547-560) and 5.2 (617-622). This makes the reading of the paper very tedious.

Third, the English used in the paper needs significant improvements and corrections. Sometimes, it is very difficult to understand what the authors really want to say. For example, the expression "the effect mechanism of (. . .) meteorological conditions" or the "effect mechanism of pollutant transport" is central to the paper and appears five times. However, it isn't clear what the authors mean with the "effect mechanism".

Other major comments:

In the abstract and at several other places in the document, the authors mention the pollution transport pathways through "Nanxiang Basin-Yunmeng Plain pathway and the Dabie Mountain's Hilly Area-Yunmeng Plain pathway". If you are not from China, it is quite hard to understand these descriptions unless you provide a map.

Lines 126 – 129: The authors argue that their findings are of practical value by broadening the scientific understanding of heavy pollution formation mechanisms. However, the findings may be of scientific interest but I cannot see the practical value. The differences to other regions of China (and perhaps to other regions with similar topography) are not much discussed in the paper. In addition, because the authors themselves demonstrate in section 5.2 that chemistry transport models like WRF-CHEM are able to reproduce the pollution transport towards the Hunan and Hubei provinces, the question arises which additional insights this study gives.

Section 2.2: The methods are not well described, the explanations are too brief and too general. The authors should have made an attempt to better explain their approach to the reader. They could also give references for those who are not deeply involved in MV-EOF and EEOF or give more details in an appendix.

Line 183: It is not clear what "synthetic analysis" exactly means in this context.

Line 221: What is meant with the "most obvious spatial characteristics"?

Figure 2: What re the units of the color scale? If this is PM2.5 is this in $\mu$g/m$^3$ or perhaps mg/m$^3$? What is the unit of the arrow? The number 0.15 below the arrow doesn't have units, either.

Table 1: You list 8 typical regional pollutant transport events. No 2 & 3 as well as 6 & 7 have just one day difference. How can you consider them as another event? In the rightmost column it is stated that they belong to the same transport event.

Figure 3: The arrows in (c) and (d) have different standard length (6 in (c) and 3 in (d)) which is misleading

Line 308 – 310: It is not clear what you want to say with the temperature differences. A change by 3$^\circ$ or only 1$^\circ$ in 24h does not seem to be much.

Line 330: What is Figure 2d?

Figure 5: If boundary layer height is given in color scale, what are the units and what

does this figure say?

Figure 6: You should not use different standard lengths for arrows when graphs are combined in one figure.

Line 374: What would you call a "pollution-free day"? This might be very subjective.

Line 407: How was the synthesis of the 8 pollutant transport events done?

Figure 9 and Figure 10: How were the soundings "synthesized"? Is this an average of the vertical soundings? If yes, this could be misleading.

Line 493: What is meant with "cold air degenerates"?

Lines 579 – 585: It is quite hard to understand what is meant in this sentence.

Lines 586 – 597: Even if the WRF setup that was used is described somewhere else, it needs a better and more detailed explanation here, e.g. what was the spatial resolution and the number of vertical layers?

Line 726 - 731: I could imagine that there are AQ forecast systems running in China and such a "predictive warning sign" is not much needed.

---

## Referee Comment (RC2) · Anonymous Referee #2 · 10 Mar 2021

Title: Meteorological formation mechanism of regional transport in winter heavy air pollution events in the middle Yangtze River area, China

The authors have carried out MV-EOF and EEOF analysis to select and understand the peak pollution episodes and corresponding pollution pathways and meteorological conditions during these events using data from 2015 to 2019 over the middle Yangtze River area in China. In addition, they have also carried our chemistry-climate model simulations to understand the same issue but for a typical event for corroboration.

Though I would like to appreciate the overall effort and the intention of the authors, I found it hard to follow the manuscript due to the following reasons.

[Figure]

The weakest part of the paper is the data and methodology section where scant information is provided with regard to the data, its curation and analysis including the details about numerical simulations.

The whole paper depends on the analysis of PM2.5 pollution based on data from the China's National Ambient Air Quality Monitoring Network. Even basic information about this dataset is missing in the manuscript. For example, it is not clear whether this is a gridded data or station data? What is the temporal resolution? What is the spatial resolution of the chemistry-climate model simulations?

Since the objective of the paper is to study the peak pollution episodes during the five-year period during the winter/January, which mode of the EOF was used finally used to create the several figures in the results section?

Was the climatology of the five-year period removed during the analysis? If not, the first mode will show up the climatological mean as the dominant feature.

Line 184 to 186 states about some synthetic and correlation analysis including anomalies. Was this the base database used for further EOF analysis?

Though the authors use several datasets and tools from the surface, reanalysis, and model simulations, the lack of information above basic aspects does not allow me to be positive. As such, in the current form, this manuscript requires substantial revision in terms of its readability and usefulness for a wide range of audiences of this journal. Hence I recommend rejection of the manuscript.

Minor Comments 1. Line 62 to 66, The sentence may be shortened.

2. Line 73, in in is repeated

3. Line 76, What is meant by excessive anthropogenic emissions? Is there any specific emission relevant only for winter that does not exist during other periods?

4. Most sentences are excessively long to understand. An English correction may help

improve the readability of the manuscript.

5. The authors need to explain where the study region is using a map. The area shown in the map is a huge region over China spread over tens of degrees across. It will be better to name the regions in a map for the reader's benefit.

6. Somehow, the periods of peak pollution are similar over the years (early and late part of January) with a bi/tri-weekly separation between them. Is there any specific reason for this?

7. In section 3, local conditions leading to high pollution are mentioned. Are these not meteorological conditions? Perhaps, it may be mentioned as local and regional or large-scale meteorology.

8. In line 215, there is a mention of the use of data from 31 urban monitoring stations. It will be better if a table is provided with all datasets used in the study with their source, frequency, and time periods.

9. In Figure 2b, how much of the variance is explained by the mode shown?

10. Selection of peak pollution events along with the time coefficient must be shown in figure 2 to identify the events.

11. Since the peak episodes are few in number, is it possible to show each of the episodes for their PM2.5 spatial patterns along with the circulation patterns (as sub-panels)? This will allow us to know whether the patterns are similar or dissimilar for each episode.

12. Use similar color bars and arrow lengths (Fig.3) so that comparison becomes easier.

13. Figure 4 corresponds to nationwide station data or reanalysis?

14. If showing from reanalysis, anomalies with respect the climatology will show a better pattern with slowing winds/lower or higher temperature over the large domain. It

appears 4a corresponds to actual winds and 4b corresponds to anomalies in temp or are both anomalies.

15. In Fig. 8, it is seen that the topographic features are avoided to a large extent. However, will the 1000 Mb level correspond to the surface? If possible, the temperature below the surface should be avoided when showing such plots.

16. Figure 9, sounding profiles could be shown along with climatology or the difference with respect to climatology similar to Fig. 10. This will clearly show the features during the pollution episodes. This will also validate/provide confidence in the reanalysis in case of any bias.

17. Section 5 appears to me as an avoidable addition to the overall flow of the manuscript. Even removing this section may not affect the overall discussion of the paper.

18. Section 5.2 details about WRF-Chem could be included in the data section.

19. Figures 8 and 13 could have a similar latitudinal spread so that the simulation could be compared with reanalysis easily. The simulations don't compare with reanalysis according to this figure (perhaps, due to the different time periods, but could be checked with the exact period)

20. I find that the manuscript is most China-centric with no reference to the many important and interesting similar studies carried out elsewhere. This could be included in the future for completeness.

---

## Author Comment (AC1) · 11 May 2021

Dear Editors and Reviewers,

Thank you very much for your careful review and helpful comments on our manuscript acp-2020-708. We appreciate very much your constructive comments and encouraging suggestions on our manuscript. We have accordingly made the careful revisions. The revised portions are highlighted in the revised manuscript. Please find our point to point responses to the reviewer's comments as follows:

**Responses to the reviewer 1**

*[The paper describes the meteorological processes that can lead to largely increased air pollution in the Hunan and Hubei provinces in central China in winter. The authors analyse $PM_{2.5}$ observations together with ERA interim meteorological re-analysis data by means of multivariate EOF decomposition. The paper suffers from several significant shortcomings. I do not recommend its publication in ACP.*

*First of all the analysis deals with a very regional or local effect. The authors do not make clear why it should be of interest for an international scientific audience to get insights into the meteorological effects leading to increased pollutants concentrations in the Hunan and Hubei provinces. In particular, the authors do not put their findings in the context of similar studies or similar conditions elsewhere in the world. Also, the literature cited in the paper consists almost exclusively of Chinese authors reporting about air quality in China. Of course, this is necessary in order to put the study into the context of other similar studies. However, the authors should have taken other international studies into account.]*

**Response 1:** Many thanks for referee's comments. In the revised manuscript, we have accordingly 1) clarified why it should be of interest for an international scientific audience to get insights into the meteorological effects leading to increased pollutants concentrations in the region of central China, 2) taken other international studies with literature cited in the paper into account, and 3) presented our finding about Meteorological mechanism of regional $PM_{2.5}$ transport in air pollution with an implication to environmental change as followings.

Regional transport of air pollutants from source to receptor regions is an important issue in atmospheric environment. The meteorological mechanism of regional transport has not been fully understood. The Twain-Hu Basin (THB), a sub-basin covering the lower plain in Hubei and Hunan provinces over the middle reaches of Yangtze River in central China, connects North China and East China. This basin lies in the downwind areas of heavy air pollution sources under the influence of East Asian winter monsoon, serving as a hub for the regional transport of air pollutants. The THB has become a regional heavy pollution center over recent decades (Shen et al., 2020). However, the meteorological influences on air pollutant transport with a receptor region for heavy air pollution in THB have not been systematically studied. Therefore, this study aimed to comprehensively investigate the meteorological mechanism of regional $PM_{2.5}$ transport during heavy pollution events over the THB based on the

multi-year observation to improve our understanding on environmental change.

Regional transport of air pollutants is an important issue in atmospheric environment (Mayer, 1999; Jacobson, 2001; Kim et al., 2015; Singh et al., 2017; Crippa et al., 2018). Air pollution has become a public concern on atmospheric environment (Zhao et al., 2013; Chowdhury et al., 2018, 2019; Kanawade et al., 2019). The synoptic circulations exert an important impact on air pollutant transport (Hegarty et al., 2007; Demuzere et al., 2009; Russo et al., 2014; Pope et al., 2015; Bei et al., 2016; Yue et al., 2016). Biomass burning over the source region (i.e., northern Indochina) coincided with weak westerly system over the northern South China Sea, and the aerosols were transported to downwind regions by a cold front and low-level jet (LLJ) (Huang et al., 2020b). Exports of air pollutants from the North American boundary were the result of eastward advection over the ocean and transport in a weak warm conveyor belt airflow (Owen et al., 2006). The transport of air pollutants under the control of cold front system has a significant effect on air quality (Fuelberg et al., 2007; Xu et al., 2016b; Kang et al., 2019). Good air quality often occurs under cyclonic conditions, while poor air quality is frequently associated with anticyclonic conditions (Russo et al., 2014; Pope et al., 2015; Santurtún et al., 2015). The long-range transport of polluted air masses from the North China Plain is the main factor for the sharp increases of PM2.5 concentrations in central

China (Lu et al., 2017, 2019b; Li et al., 2019b). Fine particulates can be regionally transported over a long distance with obvious trans-boundary transport, exerting an important effect on air pollution (Kim et al., 2012; Khuzestani et al., 2017; Li et al., 2019c; Yuan et al., 2019).

Regional transport of air pollutants is one of the key factors affecting air quality. The meteorological mechanism of regional transport has not been fully understood. The Twain-Hu (Hubei-Hunan) Basin (THB) in Central China is located in the wintertime downwind area of major pollution sources over North and East China under the East Asian winter monsoonal winds. To understand the meteorological mechanism of regional $PM_{2.5}$ transport on air pollution, 8 typical regional transport pollution events which occurred in THB during January 2015-2019 were selected objectively by using MV-EOF (multivariable empirical orthogonal function) decomposition with multi-source observation data, and the meteorological changes driving the regional transport of $PM_{2.5}$ in heavy air pollution in the basin was studied. The results showed that the regional transport of $PM_{2.5}$ from the source area to the THB was actuated by cold air southward invasion with anomalous northerly winds in the lower troposphere, and the vertical structure of atmospheric circulation for the regional transport is characterized with the typical pattern of southward advance of cold front with the cold air confronting the warm air mass over the THB area. In the middle troposphere existed an abnormal warm air

layer, referring as the tropospheric "warm lid", suppressing the vertical diffusion of air pollutants. With such the meteorological configurations, the warm air mass and the windward side of basin terrain forming a "barrier" of regional transport could significantly accumulate $PM_{2.5}$ for heavy air pollution in the THB, where a key receptor of $PM_{2.5}$ was built in regional transport over central and eastern China. These findings (Fig. 12) could enrich the scientific understanding of the meteorological influence on air pollution with regional transport of source-receptor air pollutants.

[Figure]

Figure 12. Diagram on meteorological mechanism of regional $PM_{2.5}$ transport with a receptor region of wintertime air pollution.

*[Second, the paper repeats its findings several times, partly even by using exactly the same expressions. The explanations given in section 4.2 are more or less repeated in 4.3 (lines 488-501), 5.1 (lines 547-560) and 5.2 (617-622). This makes the reading of the paper very tedious.]*

**Response 2:** In the revised manuscript, we have made the substantial revisions especially in English language following the referee's suggestions with modifying the repeated explanations in Sect. 4 and removing the Sect. 5.

*[Third, the English used in the paper needs significant improvements and corrections. Sometimes, it is very difficult to understand what the authors really want to say. For example, the expression "the effect mechanism of (...) meteorological conditions" or the "effect mechanism of pollutant transport" is central to the paper and appears five times. However, it isn't clear what the authors mean with the "effect mechanism".]*

**Response 3:** With the English language editing service, the language usages have been improved with correcting the errors of grammar, confusing wording and inappropriate expression in the revised manuscript. The title of manuscript, has been changed to "Meteorological mechanism of regional PM$_{2.5}$ transport in wintertime heavy pollution over the Twain-Hu Basin, Central China".

*[Other major comments:*

*In the abstract and at several other places in the document, the authors mention the pollution transport pathways through "Nanxiang Basin-Yunmeng Plain pathway and the Dabie Mountain's Hilly Area-Yunmeng Plain pathway". If you are not from China, it is quite hard to understand these descriptions unless you provide a map.]*

**Response 4:** That section 5.1"Nanxiang Basin-Yunmeng Plain pathway and the Dabie Mountain's Hilly Area-Yunmeng Plain pathway" has been deleted in the revised manuscript.

*[Lines 126 – 129: The authors argue that their findings are of practical value by broadening the scientific understanding of heavy pollution formation mechanisms. However, the findings may be of scientific interest but I cannot see the practical value. The differences to other regions of China (and perhaps to other regions with similar topography) are not much discussed in the paper. In addition, because the authors themselves demonstrate in section 5.2 that chemistry transport models like WRF-CHEM are able to reproduce the pollution transport towards the Hunan and Hubei provinces, the question arises which additional insights this study gives.]*

**Response 5:** We thank the referee for the kind suggestions. In the revised manuscript, we have made the according revisions with removing "practical value" and addressing the scientific interest of our study.

Under the stagnant meteorological conditions with weak winds, strong and thick temperature inversion layers, sinking motion and low mixing layer heights are unfavourable for the diffusion of air pollutants for the formation of heavy air pollution. Differently from air pollution in most parts of China, the pollutant transport and the meteorological mechanism in heavy pollution in the receptor region have not been systematically studied.

Furthermore, the section 5.2 about WRF-CHEM simulation has been deleted in the revised manuscript.

*[Section 2.2: The methods are not well described, the explanations are too brief and too general. The authors should have made an attempt to better explain their approach to the reader. They could also give references for those who are not deeply involved in MV-EOF and EEOF or give more details in an appendix.]*

**Response 6:** Following the referee's suggestion, we have added the brief description of MV-EOF in the revised manuscript as follows:

**2.2 Decomposition of multivariable empirical orthogonal function (MV-EOF)**

The empirical orthogonal function (EOF) analysis is a method used to identify patterns of simultaneous variation (Schepanski et al., 2016). The EOF can concentrate the information of original field into several main modes to describe the changes of the complex element field through the dimensionality reduction. The principle is to decompose the spatio-temporal matrix of observation data into a linear combination of the spatial eigenvector matrix and the corresponding time coefficient matrix.

The observation data of a certain variable field is given in the form of $X_{mn}$ matrix:

$$X = \begin{bmatrix} X_{11} & \cdots & X_{1n} \\ \vdots & & \\ X_{m1} & \cdots & X_{mn} \end{bmatrix} \tag{1}$$

where $m$ is the space point (it can be the number of stations or grid points), $n$ is the length of time series. Through EOF expansion, Formula (1) is decomposed into the product of the space function $V$ and the time function $T$, and the matrix form is

$$X = VT \tag{2}$$

where:

$$V = \begin{bmatrix} v_{11} \cdots v_{1m} \\ \vdots \\ v_{m1} \cdots v_{mm} \end{bmatrix}, \qquad T = \begin{bmatrix} t_{11} \cdots t_{1n} \\ \vdots \\ t_{m1} \cdots t_{mn} \end{bmatrix} \qquad (3)$$

$V$ is called the spatial function matrix (space mode), which represents a typical field that does not change with time; $T$ is called the time coefficient matrix, representing the weight coefficient of the spatial mode.

We process $X_{mn}$ as an anomaly, get the eigenroot $\lambda_m$ and eigenvector $v_m$ of the real symmetric matrix, and then calculate the variance contribution rate $\rho_i$ of the $i$-th eigenvector and the cumulative variance contribution rate $P_i$ of the first $p$ eigenvectors:

$$\rho_i = \lambda_i \Big/ \sum_{i=1}^{m} \lambda_i \qquad (4)$$

$$P_i = \sum_{i=1}^{p} \lambda_i \Big/ \sum_{i=1}^{m} \lambda_i \qquad (5)$$

The eigenvector represents the variation structure of a variable field, and its spatial distribution form represents the main distribution structure of variable field. The corresponding time coefficient is positive, indicating that the variable at that time has the same variation trend as this type of distribution. On the contrary, a negative coefficient denotes that the changing trend of variable at the corresponding time is opposite to this kind of distribution, and larger value means a more significant corresponding spatial distribution.

Multivariate empirical orthogonal function (MV-EOF) decomposition is an extended variant of EOF (Wang et al., 1992; 2008). In this method, two or more variables with the same time length and space points are standardized, and a new variable field is constructed, and then EOF decomposition is performed on the new variables. MV-EOF has advantages in simultaneously representing the spatial distributions of multiple elements and the spatial connections among various elements, and can be used to explore the coupling process of interactions in complex systems (Sparnocchia et al., 2003).

To obtain the synergistic variation of PM$_{2.5}$ concentration and meteorological

elements in atmospheric circulations of heavy pollution events in the THB, we choose the daily average PM$_{2.5}$ concentrations, 10-m wind speed (including meridional and zonal components) and SLP from 31 urban observation sites in the THB in January of 2015-2019 for MV-EOF decomposition. Since the magnitude of different elements varies greatly, all elements have been standardized before the MV-EOF decomposition.

The data matrix $X_{mn}$ constructed by using the four elements is as follows:

$$X_{mn} = \left[X_{m_1 n}, X_{m_2 n}, X_{m_3 n}, X_{m_4 n}\right] \tag{6}$$

where $X_{m_1 n}$, $X_{m_2 n}$, $X_{m_3 n}$ and $X_{m_4 n}$ represent PM$_{2.5}$, SLP, 10-m meridional wind and zonal wind, respectively; $m_1 = m_2 = m_3 = m_4 = 31$ is the number of urban observation sites in the THB, $n = 155$ is the length of the daily time series in January of 2015-2019; $X_{mn}$ is the extended new variable fields. Then, the new variable matrix is introduced into Formula (2) to do the EOF decomposition.

*[Line 183: It is not clear what "synthetic analysis" exactly means in this context.]*

**Response 7:** In the revised manuscript, we have changed "synthetic analysis" to "average" .

*[Line 221: What is meant with the "most obvious spatial characteristics"?]*

**Response 8:** In the revised manuscript, we have removed the "most obvious spatial characteristics" for focusing this study at the regional PM$_{2.5}$ transport.

*[Figure 2: What re the units of the color scale? If this is $PM_{2.5}$ is this in $\mu g/m^3$ or perhaps $mg/m^3$ ? What is the unit of the arrow? The number 0.15 below the arrow doesn't have units, either.]*

**Response:** In the revised manuscript, Figure 2 show the first three modes decomposed by MV-EOF with $PM_{2.5}$ loads, the SLP loads and 10-m wind loads, rather than the elements themselves. Since the magnitude of different elements varies greatly, all elements have been standardized before the MV-EOF decomposition. Therefore the elements loads are dimensionless.

*[Table 1: You list 8 typical regional pollutant transport events. No 2 & 3 as well as 6 & 7 have just one day difference. How can you consider them as another event? In the rightmost column it is stated that they belong to the same transport event.]*

**Response 9:** In the revised manuscript, we have changed to 8 typical air pollution days with regional pollutant transport.

[Figure 3: The arrows in (c) and (d) have different standard length (6 in (c) and 3 in (d)) which is misleading.]

**Response 10:** In the revised manuscript, the Figure S2 (Figure 3 of the previous version) has been corrected to the same arrow lengths.

*[Line 308 – 310: It is not clear what you want to say with the temperature differences. A change by 3 ◦ or only 1 ◦ in 24h does not seem to be much.]*

**Response 11:** In the revised manuscript, it has been corrected with "Affected by the cold air southward invasion, the 24-h temperature changes in the most northern areas of China drop by -2~ -5℃ , and decline by -2~ -3℃ in the THB."

*[Line 330: What is Figure 2d?]*

**Response 12:** In the revised manuscript, we have corrected this print error.

[Figure 5: If boundary layer height is given in color scale, what are the units and what does this figure say?]

**Response 13:** In the revised manuscript, Figure 5 show spatial distribution of correlations of daily changes of the third-mode time coefficients respectively with the SLP, atmospheric boundary layer height and 10-m wind vectors in January of 2015-2019. The corresponding discussion has been modified to "In the third mode, the heavy pollution in the THB is controlled by the bottom of the high pressure over CEC, and near-surface anomalous northerly winds as well as the upraised boundary layer (Fig. 5), which was a typical pattern of synoptic circulation for regional transport of $PM_{2.5}$ over north to central China (Yu et al. 2020). This circulation

condition could drive air pollutants from the source areas of North China to the downwind THB."

*[Figure 6: You should not use different standard lengths for arrows when graphs are combined in one figure.]*

**Response 14:** In the revised manuscript, Figure 9 (Figure 6 of the previous version) has been corrected to the same lengths for arrows.

*[Line 374: What would you call a "pollution-free day"? This might be very subjective.]*

**Response 15:** In the revised manuscript, we have removed the unnecessary descriptions of "pollution-free day".

*[Line 407: How was the synthesis of the 8 pollutant transport events done?]*

**Response 16:** In the revised manuscript, it has been corrected with "By comparing the 8-case averages with the January averages during 2015-2019, the anomalies of air pollutants and meteorology in $PM_{2.5}$ heavy pollution in the THB with regional transport of air pollutants (or transport-type $PM_{2.5}$ heavy pollution) were assessed."

*[Figure 9 and Figure 10: How were the soundings "synthesized"? Is this an average of the vertical soundings? If yes, this could be misleading.]*

**Response 17:** In the revised manuscript, we have changed "synthetic analysis" to "average". It has been corrected with "All the profiles are averaged for 8 days of transport-type $PM_{2.5}$ heavy pollution (Table 1) with the anomalies relative to the monthly mean in January of 2015-2019.

*[Line 493: What is meant with "cold air degenerates"?]*

**Response 18:** we have deleted the "cold air degenerates" in the revised manuscript.

*[Lines 579 – 585: It is quite hard to understand what is meant in this sentence.]*

**Response 19:** In the revised manuscript, we have removed all the descriptions and discussions about the WRF-Chem simulation.

*[Lines 586 – 597: Even if the WRF setup that was used is described somewhere else, it needs a better and more detailed explanation here, e.g. what was the spatial resolution and the number of vertical layers?]*

**Response 20:** In the revised manuscript, we have removed all the descriptions and discussions about the WRF-Chem simulation.

*[Line 726 - 731: I could imagine that there are AQ forecast systems running in China and such a "predictive warning sign" is not much needed.]*

**Response 21:** In the revised manuscript, we have removed the unnecessary descriptions of "predictive warning sign".

---

## Author Comment (AC2) · 11 May 2021

Dear Editors and Reviewers,

Thank you very much for your careful review and helpful comments on our manuscript acp-2020-708. We appreciate very much your constructive comments and suggestions on our manuscript. We have accordingly made the careful and substantial revisions. The revised portions are highlighted in the revised manuscript. Please find our point to point responses to the reviewer's comments as follows:

**Responses to the reviewer 2**

*[The authors have carried out MV-EOF and EEOF analysis to select and understand the peak pollution episodes and corresponding pollution pathways and meteorological conditions during these events using data from 2015 to 2019 over the middle Yangtze River area in China. In addition, they have also carried our chemistry-climate model simulations to understand the same issue but for a typical event for corroboration.*

*Though I would like to appreciate the overall effort and the intention of the authors, I found it hard to follow the manuscript due to the following reasons.]*

**Response 1:** Thank the referee very much for the careful review and encouraging comments on our manuscript. We have accordingly made the careful revisions. The revised portions are highlighted in the revised manuscript. In the following we quoted each review question in the square brackets and added our response after each paragraph.

*[The weakest part of the paper is the data and methodology section where scant information is provided with regard to the data, its curation and analysis including the details about numerical simulations.*

*The whole paper depends on the analysis of $PM_{2.5}$ pollution based on data from the China's National Ambient Air Quality Monitoring Network. Even basic information about this dataset is missing in the manuscript. For example, it is not clear whether this is a gridded data or station data? What is the temporal resolution? What is the spatial resolution of the chemistry-climate model simulations? ]*

**Response 2:** In response to the referee's comments, we have reorganized and modified the data and methodology section as follows:

**2 Data and methods**

**2.1 Observational data**

In this study, the daily average $PM_{2.5}$ concentrations in January over 2015-2019 are obtained from more than 1600 air quality monitoring stations in China (http://datacenter.mee.gov.cn/) to display the spatial distribution of $PM_{2.5}$ in central and eastern China. The air quality observation data are through quality control based on China's national standard of air quality observation. The $PM_{2.5}$ concentrations observed at 31 major cities in Hubei and Hunan provinces are selected to represent the air pollution levels in the Twain-Hu Basin (Fig. 1c) to construct the joint observation matrix of MV-EOF decomposition in the THB.

The meteorological observation data in January 2015-2019 are downloaded from the hourly surface data from the China Meteorological Science Data Center (http://data.cma.cn/). In this study, meteorological observations including sea level pressure (SLP), 2-m air temperature, 10-m wind speed and wind direction, are also used to construct the joint observation matrix of the MV-EOF decomposition of the THB,

The radiosonde observations of air temperature and wind speed from Wuhan and Changsha meteorological stations (Fig. 1c) are adopted as well to analyze the atmospheric thermodynamic vertical structures during wintertime and regional $PM_{2.5}$ transport cases in the THB in the recent 5 years.

In addition, in order to analyze the synoptic circulation pattern and atmospheric thermodynamic vertical structure during the wintertime air pollution in the THB with regional $PM_{2.5}$ transport, we use the ERA-interim daily reanalysis data with 0.25°×0.25° resolution (https://apps.ecmwf.int/datasets/data/interim-full-daily/), which consists the atmospheric boundary layer height, SLP, 2-m temperature, 10-m wind vector components u and v, as well as the geopotential height, air temperature, vertical speed, and wind vector components u and v at different vertical layers in January 2015-2019.

**2.2 Decomposition of multivariable empirical orthogonal function (MV-EOF)**

The empirical orthogonal function (EOF) analysis is a method used to identify patterns of simultaneous variation (Schepanski et al., 2016). The EOF can concentrate the information of original field into several main modes to describe the changes of the complex element field through the dimensionality reduction. The principle is to decompose the spatio-temporal matrix of observation data into a linear combination of the spatial eigenvector matrix and the corresponding time coefficient matrix.

The observation data of a certain variable field is given in the form of $X_{mn}$ matrix:

$$X = \begin{bmatrix} X_{11} & \cdots & X_{1n} \\ \vdots & & \\ X_{m1} & \cdots & X_{mn} \end{bmatrix} \tag{1}$$

where $m$ is the space point (it can be the number of stations or grid points), $n$ is the length of time series. Through EOF expansion, Formula (1) is decomposed into the product of the space function $V$ and the time function $T$, and the matrix form is

$$X = VT \tag{2}$$

where:

$$V = \begin{bmatrix} v_{11} & \cdots & v_{1m} \\ \vdots & & \\ v_{m1} & \cdots & v_{mm} \end{bmatrix}, \qquad T = \begin{bmatrix} t_{11} & \cdots & t_{1n} \\ \vdots & & \\ t_{m1} & \cdots & t_{mn} \end{bmatrix} \tag{3}$$

$V$ is called the spatial function matrix (space mode), which represents a typical field that does not change with time; $T$ is called the time coefficient matrix, representing the weight coefficient of the spatial mode.

We process $X_{mn}$ as an anomaly, get the eigenroot $\lambda_m$ and eigenvector $v_m$ of the real symmetric matrix, and then calculate the variance contribution rate $\rho_i$ of the $i$-th eigenvector and the cumulative variance contribution rate $P_i$ of the first $p$ eigenvectors:

$$\rho_i = \lambda_i \bigg/ \sum_{i=1}^{m} \lambda_i \tag{4}$$

$$P_i = \sum_{i=1}^{p} \lambda_i \Bigg/ \sum_{i=1}^{m} \lambda_i \qquad (5)$$

The eigenvector represents the variation structure of a variable field, and its spatial distribution form the main distribution structure of variable field. The corresponding time coefficient is positive, indicating that the variable at that time has the same variation trend as this type of distribution. On the contrary, a negative coefficient denotes that the changing trend of variable at the corresponding time is opposite to this kind of distribution, and larger value means a more significant corresponding spatial distribution.

The multivariate empirical orthogonal function (MV-EOF) decomposition is an extended variant of EOF (Wang et al., 1992; 2008). In this method, two or more variables with the same time length and space points are standardized, and a new variable field is constructed, and then EOF decomposition is performed on the new variables. MV-EOF has advantages in simultaneously representing the spatial distributions of multiple elements and the spatial connections among various elements, and can be used to explore the coupling process of interactions in complex systems (Sparnocchia et al., 2003).

To obtain the synergistic variation of PM$_{2.5}$ concentration and meteorological elements in atmospheric circulations of heavy pollution events in the THB, we choose the daily average PM$_{2.5}$ concentrations, 10-m wind speed (including meridional and zonal components) and SLP from 31 urban observation sites in the THB in January of 2015-2019 for MV-EOF decomposition. Since the magnitude of different elements varies greatly, all elements have been standardized before the MV-EOF decomposition.

The data matrix $X_{mn}$ constructed by using the four elements is as follows:

$$X_{mn} = \left[ X_{m_1 n}, X_{m_2 n}, X_{m_3 n}, X_{m_4 n} \right] \qquad (6)$$

where $X_{m_1 n}$, $X_{m_2 n}$, $X_{m_3 n}$ and $X_{m_4 n}$ represent PM$_{2.5}$, SLP, 10-m meridional wind and zonal wind, respectively; $m_1 = m_2 = m_3 = m_4 = 31$ is the number of urban observation sites in the THB, $n = 155$ is the length of the daily time series in January of 2015-2019;

$X_{mn}$ is the extended new variable fields. Then, the new variable matrix is introduced into Formula (2) to do the EOF decomposition.

*[Since the objective of the paper is to study the peak pollution episodes during the five year period during the winter/January, which mode of the EOF was used finally used to create the several figures in the results section?*

*Was the climatology of the five-year period removed during the analysis? If not, the first mode will show up the climatological mean as the dominant feature.*

*Line 184 to 186 states about some synthetic and correlation analysis including anomalies. Was this the base database used for further EOF analysis?]*

**Response 3:** First three modes of the MV-EOF were used finally used to create the several figures in the results section, and the climatology of the five-year period was removed during the analysis with MV-EOF. In response to the referee's comments, we have reorganized and modified Section 3 as follows:

**3. Results of MV-EOF decomposition and selection of typical air pollution events with regional transport**

**3.1 Analysis of MV-EOF decomposition**

Figure 2 shows the first three modes and their time series obtained by MV-EOF decomposition based on $PM_{2.5}$ concentration, 10-m wind speed (including meridional wind and zonal wind) and SLP at 31 observation sites in the THB in January of 2015-2019. The variance contribution rates of the three modes are 28.2%, 16.0%, and 12.0%, respectively, and the cumulative variance reaches 56.2%, all of which have passed the North test, indicating that the first three modes are independent of each other and can be clearly distinguished from other modes.

The spatial distributions of first three modes decomposed by MV-EOF could characterize the synergistic change of $PM_{2.5}$ concentrations with near-surface wind and

air pressure fields over the THB (Fig. 2). The first mode of positive $PM_{2.5}$ loads distribution corresponded to the negative air pressure loads and weak wind loads (Fig. 2a), and the second and third modes of positive $PM_{2.5}$ loads were distributed with the positive air pressure loads but with strong southerly and northerly wind loads (Figs. 2b-2c), reflecting the connections of regional $PM_{2.5}$ pollution over the THB with low air pressure and weak winds (Fig. 2a), with high air pressure centered over the THB and strong southerly winds (Fig. 2b), and with high air pressure in northern THB strong northerly winds driving the transport of air pollutants from North China to the THB (Fig. 2c). The significantly larger $PM_{2.5}$ loads in third mode comparing to the first and second modes (Figs. 2a, 2b and 2c) could imply an importance of regional $PM_{2.5}$ transport in air pollution in the THB.

[Figure]

**Figure 2.** (a) The first mode, (b) the second mode and (c) the third mode decomposed by MV-EOF with $PM_{2.5}$ loads (color contours), the SLP loads (black contour lines) and 10-m wind loads (vectors) as well as (d) the time coefficients PC1, PC2, and PC3 of the first three modes in January over 2015-2019.

The positive and negative time coefficients in the daily time series (Fig. 2d) showed the abnormally high and low $PM_{2.5}$ concentrations over the THB with meteorological influences on poor and good air quality. By , the third-mode time coefficients explained 48% of the total variance of the average $PM_{2.5}$ time variation in the THB (Fig. 3), which reflected an important role of regional $PM_{2.5}$ transport in increasing $PM_{2.5}$ in the THB.

[Figure]

**Figure 3.** Scatter plots of (a) The first mode, (b) the second mode and (c) the third mode time coefficients PC1, PC2, and PC3 respectively with the daily $PM_{2.5}$ concentrations averaged over the THB in January of 2015-2019.

In order to explore the synoptic circulations on heavy air pollution in the THB, we correlated the daily changes of the time coefficients of the three modes (Fig. 2d) and 850hPa geopotential heights, vertical velocity and wind of the ERA-Interim daily data over January of 2015-2019 in CEC (Fig. 4). As seen from Figure 4, the correlation coefficients between the first-mode time coefficients of THB' decomposed $PM_{2.5}$ and the 850-hPa heights were negative over the CEC, indicating that the anomalously high (low) $PM_{2.5}$ concentrations corresponded to the abnormally low (high) 850-hPa height field (Fig. 4a). The low (high) pressure center in the northwestern region of THB was conducive to the accumulation (removal) of surface $PM_{2.5}$ in the THB (Fig. 4a). The southerly winds block the vertical diffusion of pollutants, prone to local air pollutant accumulation and chemical transformation, which was similar to the heavy pollution in the Sichuan Basin of southwest China induced by the low-value system of the Southwest Vortex (Ning et al., 2018).

The correlation coefficients between the second-mode time coefficients and the 850-hPa height field were positive (Fig. 4b), suggesting that the heavy pollution in the THB was controlled with the high-pressure system with the obvious anticyclonic circulation and the significant downdrafts (the correlation coefficient of 850-hPa vertical velocity field is positive) at 850-hPa (Fig. 4b), which inhibited the vertical spread of air pollutants, strengthening the cumulative air pollution in the near-surface layer. This mechanism was also reported as a typical synoptic circulation for heavy $PM_{2.5}$ pollution in central China (Yan et al., 2021).

[Figure]

**Figure 4.** Spatial distribution of correlations of the time coefficients of (a) first mode, (b) second mode and (c) third mode respectively with the 850-hPa geopotential height

(black contour lines), vertical velocity (color contours) and winds (vectors) in January of 2015-2019. The critical correlation coefficient at the 95% significance test is 0.158.

In the third mode, the heavy pollution in the THB is controlled by the bottom of the high pressure over CEC (Fig. 4c and Fig. 5), and the obvious northeasterly airflows at 850 hPa (Fig. 4c) as well as the upraised boundary layer, and near-surface anomalous northerly winds (Fig. 5), which was a typical pattern of synoptic circulation for regional transport of $PM_{2.5}$ over north to central China (Yu et al. 2020). This circulation condition could drive air pollutants from the source areas of North China to the downwind THB. The meteorological mechanism of regional transport of air pollutants was studied in the following sections.

[Figure]

**Figure 5.** Spatial distribution of correlations of daily changes of the third-mode time coefficients respectively with the SLP (black contour lines), atmospheric boundary layer height (color contours) and 10-m wind vectors in January of 2015-2019. The critical correlation coefficient at the 95% significance test is 0.158, and the THB is roughly outlined with green lines.

*[Though the authors use several datasets and tools from the surface, reanalysis, and model simulations, the lack of information above basic aspects does not allow me to be positive. As such, in the current form, this manuscript requires substantial revision in terms of its readability and usefulness for a wide range of audiences of this journal. Hence I recommend rejection of the manuscript.]*

**Response 4:** Following the referee's comments, we have made the substantial revisions with adding the information of data and methods including description of meteorological and environmental data as well as the introduction of decomposition of multi-variable empirical orthogonal function (MV-EOF) in terms of its readability and usefulness for a wide range of audiences of this journal.

*Minor Comments:*

*[1. Line 62 to 66, The sentence may be shortened.]*

**Response 5:** In the revised manuscript, we have shortened the sentence as follows: "Regional transport of air pollutants is an important issue in atmospheric environmental prevention (Wu et al., 2013b; Owen et al., 2006; Miao et al., 2017; Kozáková et al., 2019; Lu et al., 2019a)."

*[2. Line 73, in in is repeated.]*

**Response 6:** one "in" has been removed.

*[3. Line 76, What is meant by excessive anthropogenic emissions? Is there any specific emission relevant only for winter that does not exist during other periods?]*

**Response 7:** We have deleted the sentence with "excessive emissions of air pollution" to avoid the misleading.

*[4. Most sentences are excessively long to understand. An English correction may help improve the readability of the manuscript.]*

**Response 8:** With the help of English Language editing service, the English witting errors including incorrect grammar, confusing wording and inappropriate expression have been substantially revised to improve the readability of the manuscript.

*[5. The authors need to explain where the study region is using a map. The area shown in the map is a huge region over China spread over tens of degrees across. It will be better to name the regions in a map for the reader's benefit.]*

**Response 9:** Thank for the referee's suggestion. We have accordingly outlined the study area:

[Figure]

[Figure]

**Figure 1.** (a) The distribution of surface $PM_{2.5}$ concentrations (color contours, unit: μg m$^{-3}$) and 10-m wind field (vectors, unit: m s$^{-1}$) in CEC averaged during January of 2015-2019, (b) the geographical regions of China, and (c) the geographical distribution of 31 observation sites (dots) in the THB outlined with the red line, and the color contours representing the terrain height above sea level (unit: m), the red dot for Wuhan Station, and the blue dot for Changsha Station.

*[6. Somehow, the periods of peak pollution are similar over the years (early and late part of January) with a bi/tri-weekly separation between them. Is there any specific reason for this?]*

**Response 10:** Thanks for referee's suggestion. A a biweekly separation between the periods of peak pollution in the THB could be resulted from the oscillation of cold air invasion in CEC. The atmospheric quasi-biweekly oscillation provides favorable conditions for the persistence of air pollution over the BTH region in winter. During the heavy $PM_{2.5}$ pollution events, the quasi-biweekly southerly anomalies prevail persistently over eastern China (Gao et al.,2020).

*[7. In section 3, local conditions leading to high pollution are mentioned. Are these not meteorological conditions? Perhaps, it may be mentioned as local and regional or large-scale meteorology.]*

**Response 11:** In the revised manuscript, it has been corrected with "Under stagnant meteorological conditions with local weak winds, strong and thick temperature inversion layers, sinking motion and low mixing layer heights are unfavourable for the diffusion of air pollutants for the formation of heavy air pollution."

*[8. In line 215, there is a mention of the use of data from 31 urban monitoring stations. It will be better if a table is provided with all datasets used in the study with their source, frequency, and time periods.]*

**Response 12:** In the revised manuscript, we have provided all datasets used in the study with their source, frequency, and time periods.

*[9. In Figure 2b, how much of the variance is explained by the mode shown?]*

**Response 13:** In the revised section 3.1 Analysis of modal results of MV-EOF decomposition, we have estimated the variance contribution rates of the first three modes 28.2%, 16.0%, and 12.0%, respectively, and the cumulative variance reaches 56.2%, all of which have passed the North test, indicating that the first three modes are independent of each other and can be clearly distinguished from other modes.

*[10. Selection of peak pollution events along with the time coefficient must be shown in figure 2 to identify the events.]*

**Response 14:** Following the referee's suggestion, we have added the new Fig. 6 to identify the typical events.

[Figure]

**Figure 6.** Daily changes of the standardized third-mode time coefficients PC3 in the THB in January of 2015-2019 with the red triangles representing the 8 peak pollution days with regional PM$_{2.5}$ transport selected based on the standardized PC3 exceeding 1.7 (black dot line).

*[11. Since the peak episodes are few in number, is it possible to show each of the episodes for their PM$_{2.5}$ spatial patterns along with the circulation patterns (as subpanels)? This will allow us to know whether the patterns are similar or dissimilar for each episode.]*

**Response 15:** Following the referee's suggestion, Figure S1 showed the spatial distributions of PM$_{2.5}$ concentration and 10-m wind field at the 8 peak pollution days with regional PM$_{2.5}$ transport over CEC.

[Figure]

[Figure]

**Figure S1.** Spatial distribution of daily mean PM$_{2.5}$ concentrations (unit: μg m$^{-3}$) and 10m wind vectors (unit: m s$^{-1}$) in central and eastern China (CEC) at 8 typical regional PM$_{2.5}$ transport days in the THB, which is roughly outlined with the black lines.

*[12. Use similar color bars and arrow lengths (Fig.3) so that comparison becomes easier.]*

**Response 16:** In the revised manuscript, we have redrawn the Figures with similar color bars and same arrow lengths.

*[13. Figure 4 corresponds to nationwide station data or reanalysis?]*

**Response 17:** The revised Figure 8 (Figure 4 of the previous version) corresponded to nationwide station data.

*[14. If showing from reanalysis, anomalies with respect the climatology will show a better pattern with slowing winds/lower or higher temperature over the large domain. It appears 4a corresponds to actual winds and 4b corresponds to anomalies in temp or are both anomalies.]*

**Response 18:** The Figure 8 (Figure 4 in the previous version) corresponded to nationwide station data.

*[ 15. In Fig. 8, it is seen that the topographic features are avoided to a large extent. However, will the 1000 Mb level correspond to the surface? If possible, the temperature below the surface should be avoided when showing such plots.]*

**Response 19:** In the revised manuscript, we have accordingly modified the Figure as followings:

[Figure]

**Figure 10.** The meridional vertical cross-section (averaged over 112.25°E -113°E) of wind streamlines with vertical components multiplied by 10, anomalies of air

temperature (color contours, unit:℃) and wind speed (green contour lines, unit: m s$^{-1}$).
The wind streamlines are averaged for 8 days of transport-type PM$_{2.5}$ heavy pollution
(Table 1) and the anomalies of air temperature and wind speed are calculated with the
differences between the 8-day averages of transport-type PM$_{2.5}$ heavy pollution (Table
1) and the monthly mean in January of 2015-2019 based on the ERA-Interim daily data,
and the THB topography is marked with the black shadow.

*[ 16. Figure 9, sounding profiles could be shown along with climatology or the
difference with respect to climatology similar to Fig. 10. This will clearly show the
features during the pollution episodes. This will also validate/provide confidence in the
reanalysis in case of any bias.]*

**Response 20:** Following the referee's suggestion, we have modified the Figure as
follows:

[Figure]

**Figure 11.** Vertical profiles of air temperature (dot lines) and anomalies (triangle lines)
from sounding radiosonde observations at (a) Wuhan Station and (b) Changsha Station.
All the profiles are averaged for 8 days of transport-type PM$_{2.5}$ peak pollution (Table 1)
with the anomalies relative to the monthly mean in January of 2015-2019.

[Figure]

**Figure S3.** Vertical profiles of wind speed (dot lines) and anomalies (triangle lines) from sounding radiosonde observations at (a) Wuhan Station and (b) Changsha Station. All the profiles are averaged for 8 days of transport-type PM$_{2.5}$ peak pollution (Table 1) with the anomalies relative to the monthly mean in January of 2015-2019.

*[ 17. Section 5 appears to me as an avoidable addition to the overall flow of the manuscript. Even removing this section may not affect the overall discussion of the paper.]*

**Response 21:** Following the referee's suggestion, we have removed this section in the revised manuscript.

*[ 18. Section 5.2 details about WRF-Chem could be included in the data section.]*

**Response 22:** We have removed this sections about *WRF-Chem* in the revised manuscript.

*[ 19. Figures 8 and 13 could have a similar latitudinal spread so that the simulation could be compared with reanalysis easily. The simulations don't compare with*

*reanalysis according to this figure (perhaps, due to the different time periods, but could be checked with the exact period)]*

**Response 23:** In the revised manuscript, we deleted the figures following the referee's comments.

*[20. I find that the manuscript is most China-centric with no reference to the many important and interesting similar studies carried out elsewhere. This could be included in the future for completeness.]*

**Response 24:** Many thanks for referee's comments. In the revised manuscript, we have accordingly taken other international studies with literature cited in the paper into account as follows:

Regional transport of air pollutants is an important issue in atmospheric environment (Mayer, 1999; Jacobson, 2001; Kim et al., 2015; Singh et al., 2017; Crippa et al., 2018). Air pollution has become a public concern on atmospheric environment (Zhao et al., 2013; Chowdhury et al., 2018, 2019; Kanawade et al., 2019). The synoptic circulations exert an important impact on air pollutant transport (Hegarty et al., 2007; Demuzere et al., 2009; Russo et al., 2014; Pope et al., 2015; Bei et al., 2016; Yue et al., 2016). Biomass burning over the source region (i.e., northern Indochina) coincided with weak westerly system over the northern South China Sea, and the aerosols were transported to downwind regions by a cold front and low-level jet (LLJ) (Huang et al., 2020b). Exports of air pollutants from the North American boundary were the result of eastward advection over the ocean and transport in a weak warm conveyor belt airflow (Owen et al., 2006). The transport of air pollutants under the control of cold front system has a significant effect on air quality (Fuelberg et al., 2007; Xu et al., 2016b; Kang et al., 2019). Good air quality often occurs under cyclonic conditions, while poor air quality is frequently associated with anticyclonic conditions (Russo et al., 2014; Pope et al., 2015; Santurtún et al., 2015). The long-range transport of polluted air

masses from the North China Plain is the main factor for the sharp increases of $PM_{2.5}$ concentrations in central China (Lu et al., 2017, 2019b; Li et al., 2019b). Fine particulates can be regionally transported over a long distance with obvious trans-boundary transport, exerting an important effect on air pollution (Kim et al., 2012; Khuzestani et al., 2017; Li et al., 2019c; Yuan et al., 2019).